

# An overview of sedimentary volcanism on Mars

Petr Brož[1], Dorothy Oehler[2], Adriano Mazzini[3], Ernst Hauber[4], Goro Komatsu[5], Giuseppe Etiope[6], Vojtěch Cuřín[7]

[1]Institute of Geophysics of the Czech Academy of Sciences, Boční II/1401, 141 31 Prague, Czech Republic

[2]Planetary Science Institute, Tucson, AZ, United States

[3]Centre for Earth Evolution and Dynamics, University of Oslo, Norway

[4]Institute of Planetary Research, DLR, Rutherfordstr. 2, 12489, Berlin, Germany

[5]International Research School of Planetary Sciences, Università d'Annunzio, Viale Pindaro 42, 65127 Pescara, Italy

[6]Istituto Nazionale di Geofisica e Vulcanologia, Sezione Roma 2, Italy; Faculty of Environmental Science and Engineering,

Babes Bolyai University, Cluj-Napoca, Romania

[7]Faculty of Environmental Sciences, Czech University of Life Sciences Prague, Czech Republic

*Correspondence to*: Petr Brož (petr.broz@ig.cas.cz)





**Abstract**

Extensive fields of sub-kilometre-to kilometre-scale mounds, cones, domes, shields, and flow-like edifices cover large parts
of the martian lowlands. These features have been compared to structures on Earth produced by sedimentary volcanism – a
process that involves subsurface sediment/fluid mobilization and commonly releases methane to the atmosphere. It was
proposed that such processes might help to explain the presence of methane in martian atmosphere and may also have produced
habitable, subsurface settings of potential astrobiological relevance. However, it remains unclear whether sedimentary

volcanism on Earth and Mars share genetic similarities; hence whether methane, or other gases were released on Mars during
this process. The aim of this review is to summarize the current knowledge about mud-volcano-like structures on Mars, address
the critical aspects of this process, identify key open questions, and point to areas where further research is needed to
understand this phenomenon and its importance for the Red Planet's geological evolution. We show here that after several
decades of exploration, the amount of evidence supporting martian sedimentary volcanism has increased significantly, but as

critical ground truth is still lacking, alternative explanations cannot always be ruled out. We also highlight that the lower
gravity and temperatures on Mars compared to Earth control the dynamics of clastic eruptions as well as surface emplacement
mechanism and resulting morphologies of erupted material. This implies that shapes and triggering mechanisms of mud-
volcano-like structures may be different from those observed on Earth. Therefore, comparative studies should be done with
caution. To provide a better understanding of the significance of these abundant features on Mars, we argue for follow-up

studies targeting putative sedimentary volcanic features identified on the planet's surface and, if possible, for in situ
investigations by landed missions such as that currently in progress by the Zhurong rover.

# 1 Introduction

The buoyant ascent of liquefied, fluid-rich and fine-grained sediments through a lithologic succession and its subsequent
intrusion or extrusion (hereafter referred to as mud volcanism; Kopf, 2002) is a common phenomenon on Earth. Also known

as subsurface sediment and fluid mobilization (van Rensbergen et al., 2003), it is observed in sedimentary basins typically
characterised by the rapid accumulation of fine-grained and organic-rich deposits (Mazzini and Etiope, 2017 and references
therein). Images of the martian surface acquired in the early 1970's by the Mariner 9 and Viking Orbiter missions revealed the
existence of large outflow channels likely incised by flooding events capable of transporting large amount of sediments (e.g.,
Baker and Milton, 1974; Komatsu and Baker, 1997) that were subsequently deposited in giant impact basins acting as local

depocenters (Lucchitta et al., 1986). Despite these observations, mud volcanism was never considered as a significant process
that could have shaped the surface of Mars. Only some early works hypothesised this type of activity based on low-resolution
imagery at specific localities (e.g., Davis and Tanaka, 1995; Tanaka, 1997; Ori et al., 2000, 2001). A renewed interest for
subsurface sediment mobilization processes emerged when higher-resolution images became available through the Mars
Global Surveyor (MGS), Mars Express (MEX), and Mars Reconnaissance Orbiter (MRO) missions and, importantly, some

later studies reporting the detection of methane in the martian atmosphere (Krasnopolsky et al., 2004; Formisano et al., 2004;





Mumma et al., 2009; Webster et al., 2018). On Earth mud volcanism is an important source of methane to the atmosphere (e.g., Dimitrov, 2003; Milkov et al., 2003; Etiope and Milkov, 2004). By analogy it was then hypothesised that gas released at mud-volcano-like features could be responsible for the atmospheric methane observed on Mars (e.g., Skinner and Mazzini, 2009; Komatsu et al., 2011; Etiope et al., 2011; Oehler and Etiope, 2017). Whereas today the existence of methane on Mars is debated (e.g., Oehler and Etiope, 2021; Grenfell et al., 2022) after the ExoMars Trace Gas Orbiter (TGO) failed to detect it (Korablev et al., 2019; Knutsen et al., 2021), the interest in mud volcano-like structures on Mars remains high. This geological phenomenon may have an important role in atmosphere gas emissions. Additionally, the eruption of deeply buried deposits at putative clastic eruptions sites provides a window into the subsurface and the sedimentary history of Mars which is otherwise only provided by impact crater excavation (e.g., crater central uplifts; Cockell and Barlow, 2002; Mustard et al., 2009; Quantin et al., 2012;). The source region of remobilised sediments and fluids may have been located in habitable deep environments (e.g., Michalski et al., 2013; Stamenković et al., 2021; see the review by Cockell [2014] for limitations of subsurface habitat niches). Therefore, erupted materials represent prime targets for in-situ investigations and the search for biosignatures (Westall et al., 2015).

The first hints at possible mud volcano-like structures on Mars were provided by Viking Orbiter images which revealed the presence of dozens to thousands of small pitted cones on the floors of large ancient impact basins (e.g., Allen, 1979; Frey et al., 1979). After several decades of Mars exploration, the amount of evidence supporting sedimentary volcanism has significantly increased (Fig. 1). The aim of this review is to summarize the current knowledge, identify key open questions, and point to areas where further research is needed to understand this phenomenon and its importance for Mars' geological evolution. Section 2 provides the definition of mud volcanism and review its main characteristics observed on Earth. Here, we also identify some key requirements needed to generate terrestrial mud volcanism and, by analogy, some key questions that should be addressed when searching for evidences to validate the presence of possible subsurface sediment mobilization on the Red Planet. In Section 3, we describe the morphological and morphometrical properties of potential mud volcano-like surface structures and show some typical examples for the different morphological classes, highlighting the similarities and dissimilarities to terrestrial mud volcanoes. Section 4 defines the specific martian environmental conditions and their influence on potential sedimentary volcanic processes. Section 5 assess the prerequisites and the possible timing for the occurrence of sedimentary volcanism during the martian history. In Section 6 we discuss putative sedimentary volcanism in the context of Mars` geologic evolution and its implications for habitability. Finally, in Section 7 we also suggest measurements by present and future missions (including promising sampling locations) as well as laboratory experiments and modelling activities to test the sedimentary volcanism hypothesis on Mars.

Finally we note that the authors of the paper have different opinions and views about the subject of this paper. So several aspects addressed in the work are not unanimously agreed, and therefore the paper represents an arena of debate, rather than a consensual overview.



## 2 Surface mud-fluid manifestations on Earth: definitions, origin, and variants

In this section, we briefly illustrate the main features of surface sedimentary mud-fluid manifestations observed on Earth,
which include sedimentary volcanoes (or mud volcanoes) and some variants. We also emphasize on the issue of the potentially
ambiguous identification of the various surface manifestations and terminology reported in the literature. A correct definition
of the process and related surface manifestations is essential for studying and understanding the putative sedimentary
volcanism and mud volcano-like structures identified on Mars. Once the minimum requirements, conditions and factors,
necessary to generate terrestrial mud volcanoes are defined, by analogy the same prerequisite can be inferred for the martian
structures discussed in this work, assuming these are truly a product of sedimentary volcanism. More specifically, we will try
to answer two important questions: (a) is the surface mud always stemming from a deep shale mobilised (diapir) along a fault?
and (b) is the presence of gas necessary in sedimentary volcanism? Is that gas mostly methane, like on Earth? If the answers
are yes, the martian mud-volcano-like structures are very likely sites where biomarkers can be better preserved and are, or
have been, sites of methane release to the martian atmosphere.

### 2.1 Sedimentary volcanism: definitions and genetic process

Terrestrial sedimentary volcanoes are widely studied from geological, geophysical and geochemical points of view; for the
details on their geographic distribution, inventories, characteristics, and impact on the environment, atmosphere and energy
resource exploration, the reader may primarily refer to the main review works and references therein (e.g., Dimitrov, 2002;
Kopf, 2002; Mazzini and Etiope, 2017).

However, it is crucial to clarify that not all muddy-fluid manifestations are mud volcanoes. The term "mud volcano" should
not be used for any gas manifestation resembling a mud pool or where extrusive mud gives rise to small conic edifices. Many
$CO_2$-vents, related to geothermal or hydrothermal environments, may show such characteristics. It is not only a problem of
semantics, because the attribution of "mud volcano" to a surface gas manifestation implies the existence of a series of specific
geologic processes and features.

The definition of mud volcano is strictly linked to the genetic mechanisms that are essential to generate this specific geological
phenomenon. The processes are schematically illustrated in Fig. 2. Basically, a mud volcano is characterised by the surface
release of mostly deep-seated shale deposits that are brecciated and transported along a fracture system or fault (the shale is
for a mud volcano what magma is for an igneous volcano). The shale and other rock types are fractured and mobilised, in a
diapir style, due to the combination of three factors: (a) lateral tectonic compression; (b) gravitational instability of shale, i.e.,
typically when shale is a low density sediment underlying another sedimentary rock with higher density; (c) overpressure of
gas and water from an underlying source. This source is generally a gas reservoir, which provides the necessary pressure, likely
the factor triggering the mobilization of the shale (Mazzini and Etiope, 2017). The erupted sediments are defined with the term
"mud breccia" and consist of a mix of ground-up and brecciated sediments and rocks that originate from various units and
formations intersected by the conduit. Two components are typically identified in the mud breccia a) a fine-grained matrix



(consisting mostly of clay and fine sediments in large part originating from the low density shales) and b) a set of rock clasts with sizes ranging from a few cm to cubic metres stripped from the various sedimentary formations that have been pierced by the diapir. The depth of the reservoirs (which can be found by seismic images and/or drillings) and the origin of the clasts are fundamental to assess the minimum depth of the mud volcano system. Mud volcanoes are located almost exclusively in convergent basins but they can occur along any type of fault, with normal, reverse or strike-slip kinematics (Ciotoli et al.

115     2020).

    Sedimentary volcanism on Earth is exclusively observed in petroleum-bearing sedimentary basins, i.e., in areas with gas-oil systems (source rocks, reservoirs, generally in faulted anticlines). Therefore, the erupted gas is mainly methane, associated with other hydrocarbon gases in trace amounts (ethane, propane, butane) and non-hydrocarbon gases typically occurring in sedimentary reservoirs ($CO_2$, $N_2$, minor amounts of He, and $H_2S$). Methane is mostly of deep thermogenic origin, but often

including shallower components of secondary microbial methane (Etiope et al., 2009). A few mud volcanoes in petroliferous basins may release more $N_2$, due to differential solubility in uplifted basins, which also induces higher helium concentrations (Etiope et al., 2011). Water is typically salty and enriched in various elements, largely sourced from the brines accumulated in the gas-oil reservoir (fossil water) in addition to connate water (trapped in sediments), which may also derive from illitization of clayey minerals (smectite-illite transformation; Mazzini and Etiope, 2017). Meteoric water can also mix with the fossil

water during its upwelling.

### 2.2 Variants

    The mud volcano system depicted in Fig. 2 may be more complex and a few variants of this classic scheme may exist. In some cases, the gas may stem from multilayer reservoirs and the mobilised shales may originate from different sedimentary units and, likewise, multiple source rocks may be involved in the plumbing system (e.g. Guliyev and Feizullayev, 1997; Inan et al.,

1997; Cooper et al., 2001). In addition, a set of other surface sedimentary mud-fluid manifestations exists on Earth that may have sizes and morphologies that resemble those of mud volcanoes. For this reason, the attribution of sedimentary volcanism for these variants may be uncertain, ambiguous, or susceptible to subjective and/or misleading interpretations. However, more detailed studies reveal that these structures have genetic mechanisms, and accordingly names, that are different from those of sedimentary volcanism. For example:

(a) *Sediment-Hosted Geothermal Systems*. Some sedimentary basins, featuring petroleum systems, developed above deep geothermal systems (e.g. along grabens), which can be associated with igneous intrusions, deep magmatic chambers, or lateral migration of hydrothermal fluids. These systems can be characterised by elevated $CO_2$ and $H_2O$ pressures. These settings are referred as "Sediment-Hosted Geothermal Systems" (SHGSs), where $CO_2$-dominated fluids migrate upward and mix with the gaseous hydrocarbons hosted in the shallower sedimentary formations (i.e., Procesi et al., 2019). SHGSs are essentially hybrid

systems and the relative fraction of the two endmembers (sedimentary $CH_4$ and geothermal $CO_2$, with related waters) may vary greatly. By definition, however, a SHGS releases on the surface a mixture with $CO_2$ concentrations >50 vol % (Procesi





et al., 2019) and it may have the features of a hydrothermal system, with large amounts of water vapour (as in the case of the Lusi system in Indonesia; Mazzini et al., 2012).

(b) *Artesian systems*. At mud spring sites, overpressured water displaces fine-grained sediments during upwelling forming
localised pools. These are generally shallow systems (tens or a few hundreds of metres deep), with surface manifestations within the range of some metres and mud flows whose extension may vary depending on the terrain morphology. These springs do not necessarily contain gas (e.g. Bristow et al., 2000). Dewatering structures like sand volcanoes may also display conical shaped features reaching > 10 m in size. These are shallow-rooted structures resulting from the remobilization of shallow non-consolidated sediments (Gill et al., 1957).

(c) *Sedimentary diapirism*. Here shales are purely driven by gravitative instability (similar to salt diapirism), without the need of gas overpressure; the movement is continuous but extremely slow and does not produce fluid eruptions or episodic activity on the surface (e.g. Bouriak et al., 2000; Henry et al., 2022; Bulanova et al., 2018).

(d) *Injected sands* are rapid phenomena mostly triggered by sand fluidization due to earthquakes, do not require gas overpressure and do not show presence of mud breccia. They may also be called sand injectites, they occur at depth and require
the fracturing of overburden sediments through which fluidised sands migrate eventually reaching the surface (i.e. extrudites) (Jolly and Lonergan, 2002; Hurst et al., 2006; Polteau et al., 2008).

(e) *Pingoes* are morphological deformation of the ground, typically associated with frozen aquifers (so they exclusively exist on Earth in high latitude regions), as well as associated with the presence of permafrost or gas hydrates; they are quite shallow and do not imply relevant transport of sediments but rather only surficial deformation. Their morphology may resemble that
of small mud volcanoes. Although gas is not necessary to form a pingo (which is essentially a hydrological phenomenon), methane can be released when pingoes are emplaced along faults or at pockmarks sites (Andreassen et al. 2017; Hodson et al. 2020).

## 2.3 Morphologies, rheology and mud flows in sedimentary volcanism

Mazzini and Etiope (2017) provide an overview of the dimensions (from metre scale up to 12 km in diameter) and
morphologies observed at numerous mud volcano sites worldwide and propose a classification identifying the main processes that control the ultimate shape of the structures and the extensions of the mud flows. These processes are directly connected with the mechanisms of eruption/erosion that characterize each site and can be applied also to the other surface mud-fluid manifestations described herein. Dynamic factors include the eruption frequency and the vigour (i.e. more explosive events vs less destructive events). The amount of gas/water/sediments-rocks released during the eruptions will affect the rheology of the
mud and, accordingly, the ultimate morphology of the flows. The local topography pre-existing the eruptions will affect the shape of the surface manifestations. Mechanical factors include a) the interaction between the erupted media (more or less viscous) and the surface (more or less erodible) hosting the flows, b) the type of erosion (operated by e.g. submarine currents, wind, rain) that is different depending of the geographical setting, or c) the type of e.g. subsidence associated with subsurface





dynamics. Different albedo observed at the mud flows can be used to infer the erosive efficiency of the erupted media (e.g.

Mazzini et al. 2021), or the chronology of the eruptive phases (Mazzini et al. 2009).

**2.4 Answering the key questions**

With the clarifications given above, and based on the numerous observations and studies on mud volcanoes worldwide, we

will now try to answer to the two key questions about sedimentary volcanism:

(a) is the surface mud always stemming from a deep shale mobilised (diapir) along a fault?

(b) Is the presence of gas necessary?

(a) On Earth, at least for the relatively large (tens to thousands of metres in diameter) structures, it has been documented that

the main mud component typically originates from deep shale units, upwelling even from depths of several kilometres, as

demonstrated by studies conducted on mud breccia deposits (e.g., Cita et al. 1981; Inan et al. 1997, Akhmanov et al. 2003).

Mud breccia and faults appear essential components of sedimentary volcanism. For small structures, a few centimetres or

metres wide, the mud can be shallower and the process may be more similar to an artesian system described above.

Accordingly, the answer would be "yes" for traditional and relatively large sedimentary volcanoes, with mud breccia

containing clasts and biomarkers transported from great depths.

(b) All non-ambiguous (and relatively large) mud volcanoes on Earth systematically release methane (and other gaseous

hydrocarbons), because they develop within petroleum systems. However, as mentioned above, some manifestations may also

release substantial amounts of $CO_2$ (hybrid SHGS) or $N_2$ (uplifted basins). Therefore, the answer to the second question is also

"yes". On Earth, sedimentary volcanism is always associated with gas emissions, as the gas is fundamental for the rapid and

episodic shale mobilization (Fig. 2). Gas may not be necessary for those peculiar structures, such as sedimentary diapirism and

injected sands or other dewatering structures, described above, whose genetic mechanisms are different. Fundamentally, they

do not have the "eruptive and fluid discharge" character that is typical of sedimentary volcanism.

In conclusion, it is evident that the size of the emission structure, the presence of gas, mud breccia and faults are relevant to

understand the genetic process driving mud-fluid manifestations. Larger (tens-hundreds metres wide) structures imply deeper

roots, and their formation needs relevant fluid pressures and mobilizations of large volumes of sediments. Smaller structures

(<a few metre wide) may represent shallower processes, possibly involving local aquifers and gas in the shallow subsoil. The

"size", "mud breccia" and "fault" factors, observable in the available images from Mars, shall be carefully considered in the

interpretation of the mud-volcano-like structures on the Red Planet, as discussed below. The release of gas can only be detected

by on-site ground measurements (rovers) or, in case of substantial gas plumes, by orbiters (Oehler and Etiope, 2021).



# 3 Observations

In this section, we briefly summarize the current knowledge about the morphology of mud-volcano-like structures on Mars.
Four individual sub-sections will focus on a) sub-kilometre to kilometre-scale circular mounds widely spread across the
northern lowlands, b) the kilometre-scale topographically positive features of various shapes and often associated with flow-
like edifices, c) kilometre-scale flows, and d) hundreds of kilometre-long flows and deposits. These divisions are made in order
to group features that bear similar morphological, morphometrical and spatial similarities, although overlapping characteristics
in some parameters often exist among these groups.

## 3.1 Morphologic expression of putative sedimentary volcanism on Mars

### 3.1.1. Sub-kilometre to kilometre-scale circular mounds

Many tens of thousands of bright, sub-kilometre- to kilometre-scale, circular mounds occur in the northern plains of Mars (Fig.
3). The largest abundances are in Acidalia and Utopia Planitiae, but additional examples occur in Chryse and Arcadia Planitiae,
Cydonia Mensae, the Isidis-Utopia overlap, Isidis Planitia, and possibly in the Scandia region (*e.g.,* Davis and Tanaka, 1995;
Tanaka, 1997, 2005; Tanaka et al., 2000, 2003, 2008; Farrand et al., 2005; Kite et al., 2007; Rodríguez et al., 2007; Skinner
and Tanaka, 2007; Allen et al., 2009, 2013; McGowan, 2009; Oehler and Allen, 2009, 2010, 2011, 2012 a, b; Skinner and
Mazzini, 2009; Komatsu, 2010; McGowan and McGill, 2010; Allen et al., 2013; Ivanov et al., 2014; Orgel et al., 2015, 2019;
Komatsu et al., 2011, 2016; Hemmi and Miyamoto, 2018; De Toffoli et al., 2019, 2021). While the great majority of these
mounds occur in the northern lowlands, possible examples have been reported from several localities in the highlands,
including craters in Arabia Terra (Pondrelli et al., 2011; Franchi et al., 2014), and a flat-floored depression in Terra Sirenum
(Hemmi and Miyamoto, 2017).

The mounds have typically been compared to a variety of terrestrial analogues (*e.g.,* rootless cones, scoria cones, tuff cones,
pingos, erosional remnants, clathrate degasification structures, and mud volcanoes), and in many cases, a mud volcano
interpretation has been deemed most consistent with the observed morphologies, geologic setting, associated flow structures,
and evidence for a fine-grained (low thermal inertia) sediment size.

The mounds in Acidalia Planitia (Fig. 3) were described in detail by Oehler and Allen (2010). In that study, the total area of
their occurrence was outlined, and in about half that area, 18,000+ examples were mapped. Based on that, ~40,000 such
features were estimated for southern Acidalia Planitia. Hemmi and Miyamoto (2018) studied 1,300 mounds in southern
Acidalia, using 40 digital elevation models (DEMs) to measure mound heights and basal diameters. These two studies provide
the following observations: The mounds in Acidalia (Fig.3) are circular to subcircular in plan view, with diameters ranging
from ~0.3 to 2.2 km (average ~0.8 km). The heights of the examples measured by Hemmi and Miyamoto (2018) average
15.2 m, with a range of 1.1 to 69.5 m. In cross-sectional profile, many of the mounds appear as domes (Fig. 3a), commonly
displaying a central depression (Figs. 3c-d), or steep-sided cones. Many are surrounded by moats (Fig. 3d). Some bright



circular mounds may appear to be nearly flat and more irregular in plan view, and these can sometimes enclose large, boulder-like knobs (Fig. 3e). Farrand et al. (2005) first recognised and termed these morphologies as domes, cones, and splotches. Most have a high albedo and smooth surfaces relative to the surrounding plains. Some have apron-like extensions of the smooth, high albedo material onto the plains. Similar smooth-textured, high-albedo material forms occasional, lobate flow-like structures that emanate from some of the mounds (Figs. 3c,f). Thermal Emission Imaging System (THEMIS) Nighttime Infrared (IR) images show the mounds to be dark in Nighttime IR (compare Figs. 3a-b), implying that their thermal inertia is lower than that of the plains and likely reflects a finer grain size than that of the plains (Oehler and Allen, 2010). Farrand et al. (2005) concluded that "dried, loosely cemented, mud deposits would be a good match to both the albedo and thermal inertia of these mounds". The mineralogy of the bright mounds in Acidalia has been investigated with data from the Compact Reconnaissance Imaging Spectrometer for Mars (CRISM), but results have not been definitive, possibly due to enhanced coatings of poorly crystalline materials (Oehler and Allen, 2010).

### 3.1.2 Kilometre-scale cones, domes and shields

While the circular mounds discussed in previous subsection show relative uniformity in their appearance within individual fields, this is not the case for members of the kilometre-scale feature type. Members of this group have been reported from Valles Marineris in Candor and Coprates Chasmata (Harrison and Chapman, 2008; Chan et al., 2010; Okubo, 2016; Kumar et al., 2019; Wheatley et al., 2019), in the southern part of Chryse Planitia (Komatsu, 2010; Komatsu et al., 2011, 2016; Brož et al., 2019), in the Amenthes/Nephentes region (Skinner and Tanaka, 2007), or in Hydraotes Chaos (Meresse et al., 2008). Edifices these regions display a wide range of shapes and sizes (Fig. 4), both among individual fields as well as when compared between them. Some edifices have a conical morphology including well-developed summit craters (Fig. 4a), others have no such craters (Harrison and Chapman, 2008; Chan et al., 2010; Okubo, 2016; Kumar et al., 2019; Wheatley et al., 2019). Other structures have the cross-sectional shape of domes with steep sides and flat summit areas that may feature small central knobs (Fig. 4b). Additional morphologies include shield- or pie-like edifices with one or multiple summit craters and well-developed lobate margins (Fig. 4c; Komatsu et al., 2016; Brož et al., 2019). The central craters associated with kilometre-scale cones and shield-like edifices might be either enclosed (Fig. 4a) or breached (Fig. 4e). In several regions, some of these edifices are associated with flow-like units over which they might be superposed (Fig. 4d; Okubo, 2016; Brož et al., 2017), or embayed by (Fig. 4e; Skinner and Tanaka, 2007; Brož and Hauber, 2013).

Edifices of this group have broad variations in their sizes, both among features in individual fields as well as among various fields. For example, the field of ~130 conical features within Coprates Chasma range in sizes from 0.2 km to 2 km in diameter, with average of 0.8 km (based on 59 edifices; Brož et al., 2017). Edifices are from 75 to 250 m high and their flanks are between 15 and 25° (based on 6 edifices; Brož et al., 2015a). On the other hand, ~170 pitted cones in Amenthes/Nephentes region are much wider than the edifices in Coprates Chasma and range from 3 to 15 km in diameter (average 7.8 km, based on measurements of 92 edifices). They are also higher, from 30 to 370 m (average ~120 m, based on measurements of 53 edifices)





and their slopes are typically below 10°. Features among this field also have much wider craters as compared to those in Coprates Chasma, and the craters are often breached. The crater floors can be below the level of the surrounding plains (Brož and Hauber, 2013). The field displaying the largest known variation among individual edifices is situated in the southern part of Chryse Planitia. Here, more than 1,300+ structures can be classified into five different types – cratered cones, shield-like

edifices, domes with a small central knob on their summit craters, irregular pies without significant topographical expression and kilometre-scale flows (Brož et al., 2019). These edifices can range in size from 0.2 km up to 20 km.

Because they have a similar variety of shapes and are commonly associated with flows, it is common for terrestrial km-scale mud volcanoes to be proposed as potential analogue (e.g., Jakubov et al., 1971; Aliyev et al., 2015; Mazzini and Etiope, 2017). However, the debate whether some of these kilometre-scale features were formed by sedimentary volcanism is not yet settled

as igneous volcanism has been invoked as an alternative mechanism for their formation in several cases (Lucchitta, 1990; Meresse et al., 2008; Brož and Hauber, 2013; Brož et al., 2015a; 2015b; 2017). If this is indeed the case, some of these features might represent martian analogues to terrestrial scoria cones, lava domes, tuff rings and/or tuff cones.

The spectral data from CRISM have been used to gain insight into their formation mechanism, however, the data did not unambiguously show the presence of phyllosilicates, carbonates, or sulphates in association with these edifices (Dapremont

and Wray, 2021). Hence, their sedimentary origin is not supported or refuted by mineralogical information. Additionally, Brož et al. (2017) reported the presence of partially dehydrated opaline silica, polyhydrated and monohydrated sulphate, high-calcium pyroxene, likely combined with olivine, and hydrous silica in the summit area of one cone, and mafic minerals associated with a flow-like unit in Coprates Chasma. These authors proposed that igneous volcanism might explain these observations, hence questioning a sedimentary origin of the field in Coprates Chasma (Okubo, 2016). To date, spectral

observations do not support a sedimentary origin over an igneous one, or vice versa.

### 3.1.3 Kilometre-scale flows

Kilometre-scale flows (KSFs) have been identified on several martian low plains; Chryse (Komatsu et al. 2011, 2016; Brož et al., 2019; 2022a), Acidalia (Ivanov et al., 2015; Ivanov and Hiesinger, 2020), Utopia (Ivanov et al., 2014, 2015) and Elysium Planitiae (Wilson and Mouginis-Mark, 2014), as well as at the highland-lowland boundary (HLB) between southern Utopia

Planitia and Terra Cimmeria (Skinner et al., 2007, 2009). They represent a morphologically diverse group of landforms characterised by a) spatially extensive spreading controlled by local topography, and b) the presence of lobate margins indicative of flow processes. In some martian volcanic provinces, like Tharsis or Elysium, such landforms have been interpreted as lava flows (e.g., Hartmann & Berman, 2000; Bleacher et al., 2007; Keszthelyi et al., 2008; Hauber et al., 2009, 2011). Based on remote sensing data, however, subsurface sediment mobilization has been proposed as the most likely

formation mechanism for some of these features (e.g. Skinner and Tanaka, 2007; Ivanov et al., 2014, 2015; Wilson & Mouginis-Mark, 2014; Komatsu et al. 2011, 2016; Okubo, 2016; Brož et al., 2019).



KSFs are often elongated in plan-map view as the material released from their source areas flowed along the local topographic gradient and/or filled nearby terrain depression(s). However, in flat areas of Chryse and Utopia Planitiae their length-to-width ratio is comparatively lower. KSFs vary greatly in areal extent and may feature individual or overlapping flow-like structures

(e.g, Komatsu et al., 2016; Brož et al., 2022). The largest examples are situated within Utopia Planitia. Here many overlapping flows spread over areas ranging in size from $10^2$ to $10^3$ km$^2$ (Cuřín et al., 2023). However, similar extents have been found for individual KSF such as, e.g., Zephyria Fluctus which has an area of 153 km$^2$ and an average thickness of 3.8 m (Wilson and Mouginis-Mark, 2014). This unique flow has been interpreted to be formed by the emplacement of mud by Wilson and Mouginis-Mark (2014), however, its igneous origin was not definitely ruled out. Individual flows in Utopia Planitia typically

cover an area of ~35 km$^2$ and have cross-sectional shapes of plateaus standing above surrounding terrains to heights of 20-30 metres (Cuřín et al., 2023). These are comparable to the examples found in Acidalia Planitia and Utopia-Cimmeria HLB (Skinner et al., 2007, 2009). The flows present in Chryse Planitia are smaller (few 10s of km$^2$; Brož et al., 2019).

Most KSFs comprise central and marginal units, each with specific morphologies depending on the specific region on Mars. In Utopia and Acidalia, KSFs are characterised by smooth central flat parts with randomly distributed rimless depressions

(Fig. 5e; Ivanov et al., 2015) and marginal units consisting of coalescing pits, distributory branching, and lobate hummocks (Fig. 5e,f). The KSFs of the Utopia-Cimmeria HLB consist of lobate material with varying thickness (Skinner et al., 2009), but the difference between their central and marginal units is not as pronounced. Zephyria Fluctus' central laminar flow unit has a unique polygonal surface texture with steep-sided margins (Fig. 5h,i; Wilson and Mouginis-Mark, 2014). The channelised flow-like features of Chryse Planitia are more complex and consist of three morphological elements: a central

depression, leveed channels, and a distal portion of the fading channel(s) (Fig. 5a; Komatsu et al., 2016; Brož et al., 2019; 2022a). Topographically low mounds inside the central depressions have been interpreted to mark the positions of feeder vents (Brož et al., 2022a). Similarly, Zephyria Fluctus seems to be sourced from a circular pond-like depression (Fig. 5, Wilson and Mouginis-Mark, 2014).

In Acidalia and Chryse Planitiae, KSFs are often associated with circular mounds. A similar spatial association is observed in

Utopia Planitia, where plateau-like outflows are often emanating from elongated ridges (Cuřín et al., 2023). In Utopia and Acidalia Planitiae, KSFs appear in proximity of craters with pancake-like ejecta, ghost craters (Ivanov et al., 2015) and giant polygons (Ivanov et al., 2014; Cuřín et al., 2023).

KSFs are discernible in THEMIS Nighttime IR imagery. Across all the mentioned locations, the central units are darker than the marginal one, but their contrast to the surrounding terrain varies. In Chryse and Utopia Planitiae, they appear darker than

the surroundings, while KSFs in Acidalia Planitia as well as Zephyria Fluctus appear brighter than the surrounding landforms. Using THEMIS Nighttime IR data, Komatsu et al. (2011) examined the material forming one of the KSFs within Chryse Planitia and the associated conical and dome-shaped edifices. They found that these features had a lower thermal inertia (i.e. potentially finer grainsize) than the surrounding units, making a composition by compact igneous rocks unlikely. However, it should be noted that if these flows were formed of unconsolidated fine-grained pyroclastic material, they would show a lower

thermal inertia as well.



Similar flows as KSFs can be also found in association in some regions of Mars with large valleys systems, e.g., like flows associated with Granicus and Tinjar Valles that are situated west of the Elysium Mons volcano. These flows have been interpreted to be mud flows, however, they likely formed due to the permafrost melting caused by the volcanic activity (Russell et al., 2003; Pedersen, 2013), not due to sedimentary volcanism. It remains unclear at the moment, if the mud was produced in the subsurface during melting, or formed on the surface by mixing of the released water with surface sediments.

## 3.2 Distribution and geologic setting

The martian mud-volcano-like surface features have been mainly observed in various parts of the northern lowlands of Mars, especially in Utopia, Chryse and Acidalia Planitiae. The circular mounds in Acidalia commonly occur at elevations below -4000 m in locations associated with giant polygons, which also are almost exclusively found in the northern plains of Mars (Oehler and Allen, 2012a; Allen et al., 2013; Moscardelli et al., 2012). The mounds occur individually, in pairs, in irregular clusters, and in chains. The chains may be relatively rectilinear in plan view, overlying troughs forming giant polygons (Gallagher et al., 2018), or they may be curvilinear, aligned along arcuate ridges in a type of landform termed "thumbprint terrain" (Guest et al., 1977). Facies analysis incorporating catchment areas and transport distances of sediment deposition from the Noachian to Hesperian fluvial systems and late Hesperian outflow floods predicts that southern Acidalia would be the depocenter for fine-grained, distal-facies muds (Rice and Edgett, 1997; Allen et al., 2013; Oehler and Allen, 2010, 2012a, b, 2021). The long transport distances from Noachian catchments in the highlands to depocenters in southern Acidalia would promote excellent grain size separation, such that distal facies sediments in southern Acidalia would be expected to contain thick accumulations of mud. This prediction is supported by Salvatore and Christensen (2014a, b) who used high resolution data sets to investigate morphology and spectral signatures and concluded that southern Acidalia is a region of extensive, fine-grained and water-saturated sedimentation, and Ivanov and Hiesinger (2020) who conducted a photogeological study along with crater-size-frequency distributions and concluded that a volatile-saturated, mud-rich unit was deposited in the southern Acidalia plains. Thumbprint terrain in two regions of Acidalia has been interpreted as the result of impact-generated tsunamis (Rodríguez, et al., 2016; Costard et al., 2017), supporting the concept of a northern ocean and water-saturated sediments in the Acidalia depocenters. The mounds that occur in curvilinear chains in this terrain have been interpreted as mud-volcano-like structures formed by rapid compaction in high energy flows caused by the tsunami-producing impacts (Di Pietro et al., 2021). After deposition of Noachian to Hesperian fluvial units within and below a) the region now covered by the Vastitas Borealis Interior and Marginal units of Tanaka et al., 2005 or b) the late Hesperian lowlands unit (lHl) of Tanaka et al., 2014, the late Hesperian, circum-Chryse outflow floods would have injected enormous quantities of sediments into the same area (Lucchitta et al., 1986; Baker et al., 2015; Alemanno et al., 2018). Both the earliest outflow sediments and the underlying, older strata would have been rapidly buried. On Earth, rapid burial of a volatile- and mud-rich section is ideal for development of subsurface overpressure and initiation of mud volcanism (Kopf, 2002). The circum-Chryse outflow sediments could have done



the same on Mars (Oehler and Allen, 2010). This process might additionally explain the approximate co-location of bright mounds and giant polygons in Acidalia (Oehler and Allen, 2012a; Allen et al., 2013; Orgel et al., 2019).

The Chryse outflow floods and sediment emplacement might be also responsible for the formation of kilometre-scale cones, 365 domes and shields situated in the southern part of Chryse Planitia, in a region near the termini of several large outflow channels, namely in the Simud, Ares and Tiu Valles (Komatsu et al., 2011, 2016; Pajola et al., 2016). Here, more than 1,300+ edifices, classified in five different types (including kilometre-sized cones, domes, shields and KSF), spread over the area of 700,000 km² (Brož et al., 2019). The distribution of these features shows that they are clustered and anticorrelated to the erosional remnants of ancient highlands, suggesting a genetic link between their distribution and the sedimentary deposits over 370 which they are superposed. Their distribution also shows that different types of features occur preferentially at specific latitudes (Fig. 5 in Brož et al., 2019), although the area of their extension often overlaps with regions populated also by other types. Such distribution patterns could be related to a model of sandar facies as suggested for this region by Rice and Edgett (1997). These authors identified three facies types (proximal, midfan, and distal facies) in a lateral sequence progressing from south to north. Most of the known features were mapped in the zone of the midfan facies, however, no clear correlation between 375 feature type and distance to outflow channel termini was found (Brož et al., 2019).

Another distinct distribution pattern was found for a field of ~170 pitted cones and 80+ smaller mounds (Skinner and Tanaka, 2007; Brož and Hauber, 2013) in the Amenthes/Nephentes region situated close to the dichotomy boundary, between the cratered highlands of Tyrrhena Terra in the south and smoother plains of Utopia Planitia in the north. Here, the edifices are aligned, from west to east, in a NW-SE and then W-E direction parallel to the southern margin of Utopia Planitia. To explain 380 such distribution, Skinner and Tanaka (2007) proposed the existence of annular ring basins in an impact tectonics scenario that would have acted as locations for sediment accumulation in southern Utopia Planitia and hence a source reservoir for sedimentary volcanism. Also field of ~130 pitted cones reported from Valles Marineris in Coprates Chasmata (Harrison and Chapman, 2008; Chan et al., 2010; Okubo, 2016; Brož et al., 2017; Kumar et al., 2019; Wheatley et al., 2019) show aligned distribution in NW-SE direction. Brož et al. (2017) proposed that such distribution is controlled by structures oriented roughly 385 parallel to the long axis of the Coprates Chasma tectonic graben.

The southwestern part of Utopia Planitia, in the region of Adamas Labyrintus, also displays evidence for possible sedimentary volcanism in the form of a field of more than 300 of KSFs. They have been firstly described by Ivanov et al. (2014) who referred to them as «etched flows». An additional mapping campaign performed by Cuřín et al. (2023) categorised these KSFs into 4 classes ('hills', 'ridges', 'plateaus', and 'complexly layered units'). Several KSFs can be also found throughout Acidalia 390 Planitia southward of Acidalia Mensae (Ivanov et al., 2015; Ivanov and Hiesinger 2020), as well as around the Utopia-Cimmeria HLB (Skinner et al., 2007, 2009); although no comprehensive inventory of their presence in these regions exists. In Elysium Planitia, the single flow of Zephyria Fluctus with a supposed sedimentary origin is present within the lower unit of the Medusae Fossae Formation (Fig. 5g,h,i; Wilson and Mouginis-Mark, 2014).

It remains unclear at which depth the source reservoir for the hypothesised sedimentary volcanoes is located. Hemmi and 395 Miyamoto (2018) estimated source depths for the mounds in Acidalia to be 110 – 850 m (if the mounds were formed





subaerially) and 30 – 450 m (if the mounds were formed sub-aqueously). Their work was based on bulk densities, fractures associated with co-located giant polygons, and an isostatic compensation model where the depth of the mud source was estimated from mound heights. De Toffoli et al. (2019) estimated source depths for mounds in Arcadia Planitia of 16 – 18 km. Their work was based on fractal analysis of the mounds to assess whether their spatial distributions were consistent with

control by underlying fractures and then on the assumption that upper cut offs determined by the fractal analysis reflect the depths of the fluid source. The orders-of-magnitude difference in estimated depths to source reservoirs of these two studies highlights uncertainties in both the approaches utilised for these assessments as well as the understanding of the origin of the bright mounds in the northern plains of Mars.

### 3.3 Ages

With the exception of really extensive flows, the age of martian mud-volcano-like structures is difficult to determine as they do not represent units of sufficient size for crater counting (e.g., Warner et al., 2015). Moreover, many of them have a relatively rugged topography with steep slopes. Hence, it is typically only possible to date spatially larger units with a known relative stratigraphic position with respect to the hypothesised mud volcanoes (i.e. either the edifices are superposed on these units or are partly buried/embayed by them) (e.g., Brož and Hauber, 2013; Brož et al., 2019). This approach enables bracketing their

ages by maximum and minimum ages, however, this approach is commonly fraught with large uncertainties.

This can be illustrated on the example of circular mounds in Acidalia. In most areas, these mounds have erupted onto the Vastitas Borealis Formation (VBF), a Late Hesperian to Early Amazonian (~3.2 to 1.75 Ga) unit interpreted as either a paleo-ocean deposit (Kreslavsky and Head, 2002) or a mixture of Noachian to Hesperian materials and local outflow-channel sediments (Tanaka et al., 2003). Since the majority of the mounds overlie the VBF, they must be younger than its Late

Hesperian to Early Amazonian age. Nevertheless, the minimum age has not been established. Some studies interpret the mounds as generally ancient features that formed on early Mars while fluids were still abundant in the shallow subsurface of the northern plains and perhaps while an ocean existed (e.g., Oehler and Allen, 2010; Oehler and Allen, 2012 a; Allen et al., 2013). This interpretation would be consistent with the interpretations of «thumbprint» terrain as a product of tsunamis. However, this view has been challenged by Rodríguez et al. (2019) who suggest a later stage of sedimentary volcanism that

postdates a possible ocean. Supporting a more recent origin of the features, De Toffoli et al. (2019) proposed that relatively small mounds (0.3 to 0.5 km in diameter) associated with thumbprint terrain in Arcadia Planitia may be young, with a "last occurrence" of ~370 Ma. That age is based on craters < 1 km in diameter, which because of their small size, could reflect a resurfacing age rather than a formation age (see Warner et al., 2015 for discussion of crater sizes and age interpretations).

Similar uncertainties are associated with the ages of kilometre-sized cones, domes, shields and KSFs. Specifically, the features

within the southern part of Chryse Planitia are spread over a unit that was significantly resurfaced 3.2 Ga years ago (Brož et al., 2019) and that experienced at least another two resurfacing events. As the features seems to be formed after these resurfacing events and they are partly covered by secondaries formed during the formation of Mojave crater, which has been



dated to 4.7 Ma (Werner et al., 2014), this gives a range in their age between 880 and 5 Ma (Brož et al., 2019). Similarly, cones within Coprates Chasma were emplaced on top of sedimentary deposits of Hesperian age (Okubo, 2016), but again the unit
has been later resurfaced and the cones formed after this resurfacing event. So they are likely Middle to Late Amazonian in age since some are superposed by a young landslide (Brož et al., 2017). However, some known edifices might be older, like pitted cones in Amenthes/Nephentes region that are more than ~2.4 Ga old (Brož and Hauber, 2013).

The ages of KSFs in Utopia and Acidalia are similarly uncertain. Ivanov et al. (2015) tied their formation to the later stages of VBF emplacement, hence their ages should be 3.57 Ga and 3.61, respectively. Zephyria Fluctus, situated within the Late
Hesperia transition unit (Tanaka et al., 2005; 2014) has the visual appearance of a very recent flow (Mouginis-Mark, 2013), but its precise age has not yet been determined.

## 4 Effect of the environment on sedimentary volcanism

Mars is a planet with very different environmental properties as compared to Earth. Both the surface gravitational acceleration and the current atmospheric pressure are lower, at 3.71 m/s$^2$ vs. 9.81 m/s$^2$ and ~600–1000 Pa vs. ~10$^5$ Pa, respectively. Today,
the range of atmospheric pressure at the surface of Mars can reach likely up to 12.4 mbar at the deepest point on the bottom of the ancient impact basin, Hellas, (Haberle et al., 2001, Wray, 2021) and can drop to 0.7 mbar on the top of the highest mountain, Olympus Mons. The surface temperature is on average -60°C, but it can range from -143°C at the poles up to +35°C in equatorial regions. Although average temperatures are far below the freezing point of water, locally higher temperatures at favourable seasons might be reached, theoretically enabling liquid water to be present on the surface today (Wallace and Sagan,
1979; Brass, 1980; Carr, 1983; Haberle et al., 2001; Hecht, 2002; Möhlmann, 2004; Kossacki et al., 2006; Bargery et al., 2010). However, such water would be likely very limited in time as the low atmospheric pressure would trigger boiling, freezing, and eventually evaporation into the atmosphere (e.g., Hecht, 2002; Wray, 2021). These processes might have inhibited the ability of water to propagate over the martian surface during most of its history.

Initial studies of mud behaviour at martian surface pressure were performed by Wilson and Mouginis-Mark (2014). The authors
proposed that the water present in the mud would be unstable and evaporate from the mud flow, ultimately removing the latent heat from the mixture. This implies that the residual water present in the mud mixture would freeze relatively quickly, in range of hours to days. Additional insight came from experimental work of Brož et al. (2020a,b), where the behaviour of low viscosity mud was experimentally studied in a low-pressure chamber partly simulating the Mars environment. Their results showed that low viscosity mud flows could actually propagate over cold (<273 K) and warm (>273 K) surfaces at martian atmospheric
pressure, but the mechanism of such propagation would be different from that observed on Earth. On Mars, mud propagating over cold surfaces would rapidly freeze on the surface of the flow due to evaporative cooling (Bargery et al., 2010) forming an icy-crust leading to propagation in a similar manner to pahoehoe lava flows observed on Earth (Hon et al., 1994). Once such an icy crust has formed, the interior of the mud flow is protected from additional evaporative cooling. As a consequence, mud remains liquid inside the crust for prolonged periods of time and propagates via mud tubes (analogous to lava tubes;



Calvari and Pinkterton, 1999). In contrast, low viscosity mud propagating over a warm surface boils and levitates above the
surface (Brož et al., 2020b). As the water content within martian mud flows might have varied, mud flows may have had
different viscosities. Brož et al. (2022b) experimentally revealed that the exposure of high viscosity mud to low atmospheric
pressure also leads to the formation of an icy crust, but also to a volume increase. This phenomenon occurs since the low
atmospheric pressure causes an instability of the water present in the mud mixture, leading to the formation of expanding

bubbles, which cannot escape from the high viscosity mud and increases the volume of the mud by up to 15%. Low-pressure
experiments hence demonstrate that the propagation and behaviour of mud on Mars indeed differs distinctly from that on Earth.
Sedimentary edifices built by mud flows on Mars are therefore expected to differ in shape and morphology from their terrestrial
counterparts (Brož et al., 2019; 2020a; 2022a).

Useful insights to understand the mechanisms of mud flows at low atmospheric pressures might come from theoretical

considerations of igneous volcanism on Mars, for which the roles of different gravity and atmospheric pressure have been
intensively studied (e.g., Dehn and Sheridan, 1990; Wilson and Head, 1994; Wilson and Head, 2004; Parfitt and Wilson, 2008;
Brož et al., 2021 and references therein). These investigations showed that a low surface pressure environment is capable of
affecting the ascent of magma in the feeding conduit (e.g., Wilson and Head, 1994; Parfitt and Wilson, 2008). In fact, the low
atmospheric pressure would favour the formation of bubbles and their growth within the ascending magma and hence reduce

its density. Such decrease would cause a larger density contrast to the surrounding rocks, hence in buoyancy, and would
ultimately increase the speed of the magmas ascent. Similarly, the rapid decompression of the mud during its ascent together
with rapid degassing might also favour its release via low energy explosive eruptions. In this case, muddy eruptions similar in
nature to Hawaiian and/or Strombolian igneous eruptions, might be more frequent on Mars than on Earth because significant
gas expansion within the final phase of its ascent might increase the ejection velocities of the expulsed mixture (Wilson and

Head, 1994).

As surface atmospheric pressure has varied on all timescales (for a recent review see Jakosky, 2021) and temperatures were
likely higher in the past (Haberle et al., 2017), martian mud flows might have been emplaced at times where the importance
of evaporative cooling may have been significantly reduced. It is reasonable to expect that the shapes of martian sedimentary
volcanoes may then vary significantly among various fields due to local environmental properties (Brož et al., 2022a; Cuřín et

al., 2023). To our knowledge, no dedicated studies exist that investigated such morphological variabilities. To perform such
study is, however, particularly challenging since the ages of possible martian sedimentary volcanoes as well as the exact
paleopressures in Mars' history are only poorly constrained (e.g., Kite et al., 2014).

In addition to low atmospheric pressure, Mars also has a lower gravity as compared to Earth, and it remains unknown how this
may affect sedimentary eruptions. Once again, a comparison to igneous volcanism can provide valuable hints. It has been

shown that the lower gravity on Mars tends to reduce the speed of lava flows and increase their thickness, as the liquid would
spread laterally to a lesser degree (Wilson and Head, 1994; Rowland et al, 2004). As a consequence, this would change the
heat loss rate of the lava flow. In a similar manner, mud flows could form thicker accumulations without losing heat. Moreover,
the low temperatures at the martian surface could also impede the ability of water to infiltrate into the subsurface (Conway et





al., 2011; McCauley et al., 2002; Pfeffer and Humphrey, 1998), limiting water loss from the mud mixture and thus maintaining the viscosity of the flow (i.e. it does not become more viscous by losing water). Finally, the low atmospheric pressure would also reduce the importance of cooling the mud flows by convection of the overlying atmosphere.

The lower gravity on Mars could also hinder the occurrence of sedimentary volcanism, as the lithostatic pressure within the crust has a different gradient to that on Earth. In other words, in order to achieve the same pressure as at a certain depth on Earth, it is necessary to be approximately three times deeper on Mars (i.e. the sedimentary basins need to be three times thicker). Therefore, from the theoretical point of view, it might be more challenging to have suitable depths of strata for sedimentary volcanism to arise on Mars compared to on Earth.

Taken together, there is currently insufficient understanding of the processes that might lead to subsurface sediment mobilization on Mars. There are also no analytical or numerical models to explain such processes, despite significant progress in analogue modelling over the last years. The main challenge is determining how the historical variations in atmospheric surface pressure and the low gravity on Mars might affect this process. Namely, for mud-volcano like edifices formed under atmospheric conditions inhibiting the presence of liquid water on the surface for prolonged period of time, it is unknown how evaporative cooling and the subsequent formation of an icy-crust on mud flows affects their rheology and their ability to spread across the surface. Also we do not know what the pressure gradient driving the flow of the mud up to the surface is, how a turbulent flow regime would affect the freezing rate of the mud, or when these mud volcano-like features formed and what the conditions were like on the surface of Mars at the time. In summary, these gaps complicate the development of numerical models that would be capable of reconstructing the formation of martian sedimentary volcanoes and would enable us to study the role of individual parameters in sedimentary volcanic processes on Mars. In turn, these shortcomings limit our ability to predict the surface morphology of the resulting sedimentary edifices, and hence what to search for.

## 5 Ideal prerequisites to form sedimentary volcanism on Mars?

From a general point of view, the initiation of sediment mobilization on Mars requires several conditions to be met: (a) the relatively quick accumulation of sufficiently thick sedimentary deposits, (b) the presence of at least some strata of fine-grained sediments and liquid water within these deposits, and (c) overpressurization of the sediments or gravitational instability as a trigger of fluid expulsion (and likely gas expulsion as well). In other words, sedimentary volcanism on Mars cannot occur anywhere, but should be restricted to specific locations where favourable conditions were present in the past.

As a first prerequisite for sediment mobilization and mud volcanism, sizable depocenters must have been available that could be filled over time with sufficiently thick volatile-rich sedimentary strata. On Earth, such areas of sediment accumulation are often linked to plate boundaries (e.g., Dickinson, 1974; Miall, 1984), whereas Mars is a one-plate planet (Solomon, 1978) and does not show evidence for plate tectonics (Tosi and Padovan, 2021). Therefore, other suitable locations with sufficient sedimentary accumulation have to be selected. The ideal candidate locations displaying such characteristics were identified by many studies and include a) ancient, large impact basins that would have acted as sinks for water and sediments that were





transported in giant flood events through outflow channels (e.g., McGill, 1989; Frey et al., 2002; Ivanov et al., 2014; Jones et al., 2016), and b) large troughs created by extensional tectonics and/or collapse (notably Valles Marineris), some of which may have once hosted lakes (e.g., Harrison and Grimm, 2004; Harrison and Chapman, 2008; Warner et al., 2013; Okubo, 2016). For example, Tewelde and Zuber (2013) proposed that 2–4 km of sediment fill accumulated in Acidalia Planitia and up

to 5 km of sediments were deposited in Utopia Planitia over time. However, at the moment, the exact sediment thicknesses and sedimentation rates in the basins remain unknown.

The second prerequisite is the infilling of such depocenters with fine-grained sediments that could be potentially mobilised. Several mechanisms have been proposed for the transport and deposition of such fine-grained sediments. For example, Okubo (2016) proposed that eolian deposition within the Hesperian epoch might have partly infilled the Candor Chaos and Coprates

Chasma regions and that these sediments might have been later buried by Middle Amazonian sediments and mobilised into large muddy laccoliths. Alternatively, Ivanov et al. (2014) proposed that fine-grained sediments might have originated from an ancient muddy ocean fed by outflow channels floods (see also Jöns, 1985). Similar outflow channel floods were also proposed as the source for fine-grained sediments within Chryse Planitia, as analogous to terrestrial outflow events (Rice and Edgett, 1997). An additional mechanism is the deposition of fine-grained volcanic ash originating from explosive volcanic

eruptions (Ivanov et al., 2012) resulting in thick and extensive pyroclastic deposits deposited over volatile-rich units (see Brož et al., 2021 and references therein). These pyroclastic deposits could then be subsequently buried by younger material.

The third prerequisite is the generation of overpressure or gravitational instability required to mobilize sediment, water and/or other volatiles. Hypotheses include a) top-down freezing of water-bearing (i.e. muddy) sediment bodies, i.e. the gradual thickening of a cryosphere (Clifford et al., 2010; Ivanov et al., 2014; Ivanov and Hiesinger, 2020) and b) rapid burial of

sedimentary strata by mass-wasting processes like lahars, landslides, impacts, etc. (e.g., Tanaka, 1999; Skinner and Tanaka, 2007; Skinner and Mazzini, 2009). Overpressure resulting from these types of processes may be enhanced by gas released if clathrates are destabilised (as could occur by uplift, sublimation of a frozen ocean, temperature changes, etc. (e.g., Oehler and Allen, 2010; Oehler and Etiope, 2017), d) seismic activity (e.g., .Skinner and Tanaka, 2007), e) rapid changes in the local tectonic regime (e.g., Hemmi and Miyamoto, 2017) or f) combinations of these mechanisms. In addition, one must consider

the possibility that large quantity of clathrates may have been destabilised, if, for example, a frozen ocean sublimated away. If that occurred, then clathrate destabilization might be a major source of overpressure and mud volcanism in some areas. However, currently, there is neither ground truth nor numerical modelling to assess such scenarios.

Despite this uncertainty it is reasonable to expect that one or more of these mechanisms were likely active in Mars` history. As individual martian sedimentary depocenters were subject to different geological settings and conditions, and the climate

and aqueous activity on Mars changed over time (cf. previous Section 4), possible sediment mobilizations did not necessarily happen at the same epochs. This could apply to both the time when the sediments were emplaced and when they were subsequently mobilised in the subsurface. It is most likely that water-rich sediments were emplaced when liquid water was more widespread on Mars than in the Middle to Late Amazonian. It seems reasonable to assume that the sediments were therefore accumulated when the valley networks and, perhaps more importantly, the outflow channels were formed (Late



Noachian-Early Hesperian, and Late Hesperian-Early Amazonian, respectively). As for the mobilization, it is also likely that most potential triggers for mobilization would have been more active in the Noachian and Hesperian than in the Middle to Late Amazonian (e.g., large impacts: Hartmann and Neukum, 2001; seismicity: Knapmeyer et al., 2008). The exception is the thickening of the cryosphere (top-down freezing), which is probably an ongoing process. As a result of such changes and the associated effects of the martian environment on mud behaviour (see previous Section 4), martian putative sedimentary

volcanoes and mud flows might show a large morphological and chronological variability. In fact, distinct depocenters would host differently-sized mud reservoirs, with sediment of diverse lithologies and geochemical provenance, contain variable amounts of available water, and experienced diverse conditions (e.g., overpressurization, density stratification) that may have triggered sediment liquefaction and mobilization. Variations within this parameter space could have controlled the effusion rates during mud expulsion as well as the water content within the mud, and in turn the viscosity of the ascending mixture

might show a broad variability (e.g., Brož et al., 2019) resulting in a potentially large morphological diversity of sedimentary landforms (see previous section).

## 6 Addressing the key questions about sedimentary volcanism for Mars

This review aims to outline the fundamental concepts and definitions of mud volcanism on Earth and to identify and describe martian examples of features that could be associated with a similar formation mechanism. The fundamental question is: Are

these martian features analogous to terrestrial mud volcanoes? Or, in contrast: Is it impossible to provide a definitive answer based on our current knowledge? Throughout this review we have applied the term «mud volcano-like» for structures that may have had a similar formation process to mud volcanoes on Earth. Considering the parameters that are commonly used to define mud volcanism-related phenomena on Earth, there are several aspects that need to be resolved and further investigated before we can apply Earth-based definitions to martian structures. Whereas the surface morphology of the investigated features is

relatively well characterised at least at the meter to kilometre scale (Section 3), the subsurface mechanisms leading to sediment mobilization as well as the link to tectonic settings remain unclear. We still cannot constrain the composition and origin of the mobilised sediments emplaced on the surface, the associated amount of water and gases, nor the triggering mechanism that would initiate sedimentary volcanism on Mars. Section 2.4 underlined key questions that should be addressed to better assess the potential for mud volcanism on Mars.

*Sedimentary deposits and structural discontinuities*. Most of the largest terrestrial mud volcanoes are formed in areas of major compressional stress at plate boundaries, and the source areas are at kilometre-scale depths (Mazzini and Etiope, 2017). It is known that thick accumulations of layered sediments exist on Mars, and they can locally reach thicknesses of several kilometres (e.g., Malin and Edgett, 2001; Milliken et al., 2010). Although these deposits (e.g., in Valles Marineris or in Gale crater) are exposed at the surface and are most likely not a viable source for deep-seated mud volcanism, several ancient

depocenters especially in large, ancient impact basins are believed to host kilometres-thick depositional units at depth (e.g., Lucchitta et al., 1986; Goldspiel and Squyres, 1991) that could act as significant reservoirs of mobile and potentially buoyant



sediments which could be ultimately erupted at the surface. At the termination of outflow channels, voluminous sediments must have been deposited by the flooding events and reach minimum thicknesses of more than several hundred metres (e.g., Tanaka, 1997). Direct observational evidence of such deeply buried sediments is lacking, however, and multispectral analyses

of materials excavated by impact crater formation in the northern lowlands do not strongly support thick deposits of sediments in the basins with the most numerous mud volcano candidates (Chryse, Acidalia, Utopia) (Pan et al., 2017). Geophysical results are inconclusive, too. Although the average bulk density of the martian crust is lower than expected than for a mafic (basaltic) crust (Goossens et al., 2017) and lateral variations may exist, the average density seems to be lower in the southern highlands than in the northern lowlands (Wieczorek et al., 2022), which appears inconsistent with large volumes of low-density sediments

in the lowlands (although it should be noted that the highest crustal densities in the lowlands correspond to volcanic provinces, not the large impact basins; Belleguic et al., 2005; Goossens et al., 2017). Nevertheless, recent seismic observations obtained through measurements by the InSight lander indicate lower shear-wave velocities in the lowlands as compared to the highlands, which could be due to thick accumulations of sediments (Li et al., 2022). New orbiters with powerful radar instruments and high-resolution gravity and measurements (Genova, 2020; Oberst et al., 2022) would be essential to identify and characterize

any voluminous wet sediments in the deep martian subsurface.

The importance of faults and other structural discontinuities to facilitate the sediment/fluid flow on Earth has been highlighted in Section 2. Mars has lacked plate tectonics for most (or possibly all) of its history (e.g., Grott et al., 2013, Smrekar et al., 2019 and references therein), and although poorly constrained, the martian subsurface is considered to be extensively faulted and fractured after a long history of impacts, the formation and evolution of the dichotomy, and the uplift of Tharsis (Golombek

and Phillips, 2010). The northern lowland areas, where most of the candidate mud volcanoes are observed, are structurally dominated by wrinkle ridges, considered to be evidence for contractional deformation of a layered substrate (see review of Mueller and Golombek, 2004). Therefore, there are certainly tectonic discontinuities in the areas of possible sedimentary volcanism. However, they are mostly indicative of compressional stress, and the dip angle of the thrust faults associated with wrinkle ridges is relatively shallow (average ~30°, according to many models; e.g., Karagoz et al., 2022). Although there may

be a large range of possible dips (see discussion by Andrews-Hanna, 2020), it is not clear whether these faults may have acted as preferential pathways for liquefied sediment.

*Hydrocarbons.* As explained in Section 2, mud volcanism on Earth is exclusively observed in hydrocarbon-bearing sedimentary basins and commonly associated with the release of significant volumes of methane (Mazzini and Etiope, 2017), a gas that significantly contributes to the greenhouse effect. At the moment, we do not know what type of gases (if any) may

have been released if subsurface sediment mobilization was ever active on Mars. A summary of current observations on that regard have been provided by Oehler and Etiope (2021). The authors discuss the potential for methane on Mars (with both abiotic and potential biotic sources assessed) as well as the discrepancies between the methane measurements by Curiosity (Webster et al., 2018) and the non-detections by the ExoMars Trace Gas Orbiter (Korablev et al., 2019). Oehler and Etiope (2021) conclude that there is a strong case for the production of abiotic methane in the subsurface of Mars. Much of that could

have been produced in the early history of the planet and may have already seeped to the surface. If so, remaining quantities





are likely to be trapped by the planet wide cryosphere, resulting in minor and episodic releases – potentially below detection limits of TGO. Nevertheless, methane may have been present in the martian subsurface in substantial amounts early in the planet's history. In addition, since the martian atmosphere is $CO_2$-rich, both methane and $CO_2$ are potential gases that could have been released with sedimentary volcanism on Mars. If we assume that martian sedimentary volcanoes were indeed

accompanied by large releases of methane (or other greenhouse gases), this would imply that those gases could have contributed to transitory climate warming in the past as well as the potential for the existence of methane reservoirs at depth (Oehler and Etiope, 2021). However, as no signs of active sedimentary volcanism on Mars have been discovered so far, and the ages of putative sedimentary edifices determined by crater counting suggest an activity at least dozens or hundreds of millions of years ago (Brož and Hauber, 2012; Brož et al., 2017; 2019), it is therefore highly unlikely that ancient sedimentary

volcanism is the source for martian atmospheric methane detected today (e.g., Giuranna et al., 2019; Oehler and Etiope, 2021 and references therein). Lefèvre and Forget (2009) showed that methane should have a relatively short lifetime in the current atmosphere of Mars – around 300 years in the upper atmosphere, but only 200 days or less when close to the surface. Therefore, recent methane detections are likely released by different mechanisms, e.g., by seepage from partly sealed subsurface reservoirs. And as such, it remains unclear at the moment if a genetic link between martian mud volcano-like structures and

methane releases exists at all.

*Triggering mechanisms:* On Earth, gravitational instability of shales and gas overpressure as well as water present at buried deposits are crucial factors to promote sediment mobilization (Mazzini and Etiope, 2017). It is extremely unlikely that all these mechanisms operated (or even operate) in the subsurface of Mars (e.g., Oehler and Allen, 2010) due to the likely absence of comparable quantities of hydrocarbons and plate tectonics. Other mechanisms (summarised in Section 5) have therefore been

proposed. For example, it was proposed by Oehler and Etiope (2021) that clathrate destabilization might account for the majority of the bright mounds in Acidalia where shallow-rooted conduits are suggested following a model similar to that recorded from, e.g., Lake Baikal (Khlystov et al., 2019). If such scenario is proven to be correct, the vast number of these mounds would reveal the extent to which clathrates are buried across the martian northern lowlands.

However, as none of these scenarios are supported by ground truth evidence nor by physical models nor by numerical

modelling results, this limits our ability to understand the way the sedimentary volcanism and the associated possible release of greenhouse gasses would affect the evolution of Mars. And hence, if sedimentary volcanism could help to alter the atmosphere of early Mars or not (Wordsworth, 2016).

## 7 Ground truthing and tests of the sedimentary volcano hypothesis

Currently available remote sensing data, despite their diversity and high spatial resolution, provide only limited insights into

sedimentary volcanism, and, in many cases, the surface morphologies can have alternative intepretations (e.g., Beven, 1996). Spectral data obtained from orbit are in conclusive (e.g., Komatsu et al., 2016; Dapremont and Wray, 2021). A definitive proof of sedimentary volcanism on Mars is hampered by the lack of *in-situ* data (ground truth) that should provide unambiguous



evidence. Until now, in situ observations supporting the presence of subsurface sediment mobilization are very limited. In this section, we describe the most promising examples that have been, or can be, visited on the martian surface, and suggest tests
of the sedimentary volcano hypothesis.

## 7.1. Sedimentary pipes in Gale crater

The Mars Science Laboratory mission with its rover, Curiosity, has traversed fluvio-lacustrine and aeolian sedimentary rocks that were deposited in Gale crater ~3.6 to 3.2 Gy ago (Rubin et al., 2017). Structures interpreted to be pipes formed by vertical movement of fluidised sediment were observed at several locations (Fig. 6a-d) (Rubin et al., 2017). Circular rings of erosion-
resistant material with diameters of 7 to 70 cm rise a few centimetres above their surroundings and display cementation and concentric internal layering. They are associated with other potential fluidised sediment features such as sedimentary dikes (Fig. 6e; Grotzinger et al., 2013) and deformational structures and may be analogues to clastic pipes on the Colorado Plateau (Ormö et al., 2004; Mahaney et al., 2004; Wheatley et al., 2016, 2019). Clastic pipes are injection features that vertically crosscut bedding with sharp contacts (Figs. 7). They display cylindrical morphologies, massive or radially graded interiors,
and raised outer rims. Increased grain size and subsequent cementation along the more porous edges makes the rims more resistant to weathering. Pipes have crosscutting relationships with other pipes due to multiple formation events or migrating eruption centres, and they are associated with other soft-sediment deformation features. Terrestrial clastic pipes form via liquefaction and fluidization, which require a near-surface groundwater system to initiate.

Another potentially interesting and relevant analogue is identified in western Japan along the Kii Peninsula coast where ancient
mud volcanism is preserved and exposed in sedimentary sequences (Komatsu et al., 2019). Coarse-grained shallow marine sediment sequences of the Miocene are intruded by the underlying fine-grained sediment. The intruding mudstone deposits exhibit diverse types of stratigraphic features, including mud dikes intruding into overlying layers or diapirs in contact with surrounding strata (Fig. 7). Like many pipes on Earth, the structures in Gale crater are more resistant to erosion than the host rock; they form near other pipes, dikes, or deformed sediment; and some contain internal concentric or eccentric layering.
These structures provide new evidence of the importance of subsurface aqueous processes in shaping the near-surface geology of Mars (Rubin et al., 2017).

## 7.2. Zhurong rover study of one putative martian sedimentary volcano

The Chinese Zhurong rover onboard Tianwen-1 landed on southern Utopia Planitia (25.066° N, 109.925° E) on 15 May 2021. In the vicinity of the landing site, orbital imagery revealed the presence of a field of cone-shaped edifices (Liu et al., 2021; Ye
et al., 2021; Zhao et al., 2021). In the field, there is one pitted cone with a height of 80 m and a basal diameter of 800 m located about 16 km southeast of the landing site, and it has attracted the attention of the mission science team (Liu et al., 2021). Alternative interpretations, including cinder cones, sedimentary volcanoes or pingos, have been proposed to explain the origin of this structure. Sedimentary volcanism appears to be the preferred origin as reported in recent studies (Ye et al., 2021; Huang et al., 2022), although other small mounds in the region have been interpreted as lava domes (Lin et al., 2023). In-situ study





690 of the closest cone to the landing site would provide a great opportunity for ground-truthing of one example of putative sedimentary volcanoes on Mars. The identification of clay minerals, like smectite, or illite (Mazzini and Etiope, 2017 and references therein), as a main bulk component of the pitted cones would be strong evidence for a mud volcano origin.

### 7.3. Future in-situ investigation of hypothesised sedimentary volcanism

Some proposals have been made regarding the importance of in-situ investigation of purported sedimentary volcanism
695 (Komatsu et al., 2014), and some candidate landing sites can be listed at some areas of hypothesised sedimentary volcanism such as those in Chryse Planitia (Rodriguez et al., 2007; Komatsu et al., 2011, 2016; Brož et al., 2019; Komatsu and Brož, 2021).

The conceivable in-situ investigations at a future landing site may include those for a) geology, b) geo-biochemistry, and c) biology. First of all, the origin of the edifices must be investigated. In-situ lithological and mineralogical examination of
700 edifices' surface or sub-surface would be essential to distinguish the formation processes. A confirmed sedimentary volcanism scenario would bring insights about the water occurrence both on the surface and in the subsurface of Mars. Further, an analysis of the erupted mud breccia clasts would provide unprecedented information regarding the subsurface geology at these sites.

On Earth mud volcanoes host a large variety of microbial communities that thrive particularly at seepage sites (e.g., Wrede et al., 2012; Kokoscha et al., 2015; Tu et al., 2017; Lee et al., 2021; Miyake et al., in press). It has even been suggested that fluids
705 ascending from deeply subducted slabs may have lead to low-temperature alteration environments in the conduits of serpentinite mud volcanoes that provided suitable niches for early life (e.g., Pons et al., 2011; Fryer et al., 2012, 2020). The study of putative mud volcano-like structures has therefore a great potential for astrobiology investigations and to collect fossilised microorganisms (e.g., Komatsu and Ori, 2000). Such investigation of martian sedimentary volcanoes should focus on localities where mud eruptions occurred, including summit craters and small mud mounds (called gryphons) where
710 emissions of mud and gas might continue even after the major eruptions for prolonged period of time. However, young fresh-looking mudflows emanating away from the summit craters are also promising candidate targets. It is recommended to conduct drilling ≥2m into the mud in order to sample materials less exposed to the harsh surface environment and protected from surface radiation (Pavlov et al., 2022). The only currently existing rover equipped to drill to that depth is the ExoMars rover (Vago et al., 2017), which is planned to land in the Oxia Planum region. While the main targets are phyllosilicate-bearing
715 layered deposits, there are some km-sized mounds in the area that form part of a regional population of mounds in the southern and eastern marginal regions of the Chryse impact basin (McNeil et al., 2021). Some of these mounds have been interpreted to be products of sedimentary volcanism (Adler et al., 2022), and if the ExoMars rover lands near one of these mounds, it is recommended that it investigates it in situ.





## 8 Conclusions

After several decades of Mars exploration, there is a growing consensus that a phenomenon similar to sedimentary volcanism on Earth may have been active on the Red Planet. An improved understanding of martian geological settings and higher quality remote sensing images have helped in assessing the origin (sedimentary versus igneous) of numerous enigmatic features observed on the surface. This manuscript reviews the martian regions were mud-volcano like features occur and provides detailed descriptions of the observed morphologies and potential formation mechanisms.

The greatest abundance of mud volcano-like features occurs in the northern lowlands of Mars, commonly north of the latitudinal maximum limit has have governed the landing-site selection for previous landed missions. The largest abundances are in Acidalia and Utopia Planitiae, where a hundred thousand of such structures occur, but examples occur also in Chryse, Arcadia and Isidis Planitiae,. Outside the northern lowlands, only a few fields of such mounds have been identified, for example in Arabia Terra, Valles Marineris or Terra Sirenum. These numerous mounds clearly reflect major events in the history of the

martian lowlands - events that could have initiated warming episodes and may even have global climatic relevance. In addition to these numerous, near-circular edifices, larger and morphologically/morphometrically more diverse features of similar origin have been also discovered. They are represented by kilometre-scale edifices with cross-sectional shapes of domes, shields, or flow-like features with a length of up to dozens of kilometres.

As outlined in this overview, there should be favourable conditions in the subsurface of Mars that enable the process of

sediment mobilization and hence sedimentary volcanism. This is because thick accumulations of layered sediments exist on this planet acting as significant reservoirs of mobile and potentially buoyant sediments and at the same time these layers might be extensively faulted and fractured enabling mobilised sediment to propagate to the surface. However, it is currently unknown what the exact mechanism responsible for such mobilization would be, because we do not know what type of gases (if any) may have been released. Therefore, whether the process of subsurface sediment mobilization on Mars is an exact analogue of

the process of terrestrial mud volcanism which is governed by gas emissions.

If the described mud-volcano like structures on Mars share a formation mechanism similar to that observed on Earth, they would represent excellent targets for future landed missions, as they could archive unaltered biosignatures, if life ever developed on Mars (Oehler and Etiope, 2021). On Earth, mud volcanoes are localities where methane is continuously released and represent ideal niches for habitability (e.g., Knittel and Boetius, 2009). They also provide a window into deep biosphere

potential sedimentary strata (e.g., Plümper et al., 2017), as mud volcanoes bring sediments from metres to kilometres of depth to the surface via a relatively low temperature and pressure processes (i.e. unlike impact ejecta that are associated with high pressure and temperatures). The study of similar features on Mars could, thus, provide insightful information about currently inaccessible buried stratigraphy.

Currently, new insights might come from the Chinese rover, Zhurong that is near a potential mud-volcano-like mound in

Utopia Planitia. Observations from this rover might provide important observations about the origin of the material(s) forming the edifice. Continuing work on analogues, using laboratory simulations of mud flows under Mars-like conditions, and



associated numerical models should help to assess the significance of some of the various edifices found on Mars. And hence this might help to predict the surface morphology of the resulting sedimentary edifices, and what to search for.

Future exploration, using drones or helicopters, may allow sample analysis or even sample collection from some of the potential sedimentary volcanoes in northern latitudes. This would then help to address many of our current knowledge gaps and provide direct insight about the mechanism of mud dynamics on Mars. The next decades of Mars exploration should therefore involve efforts to better explore these enigmatic features, which could be excellent candidates for future missions aimed at biosignature detection (Oehler and Etiope, 2021).

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





**Figures**

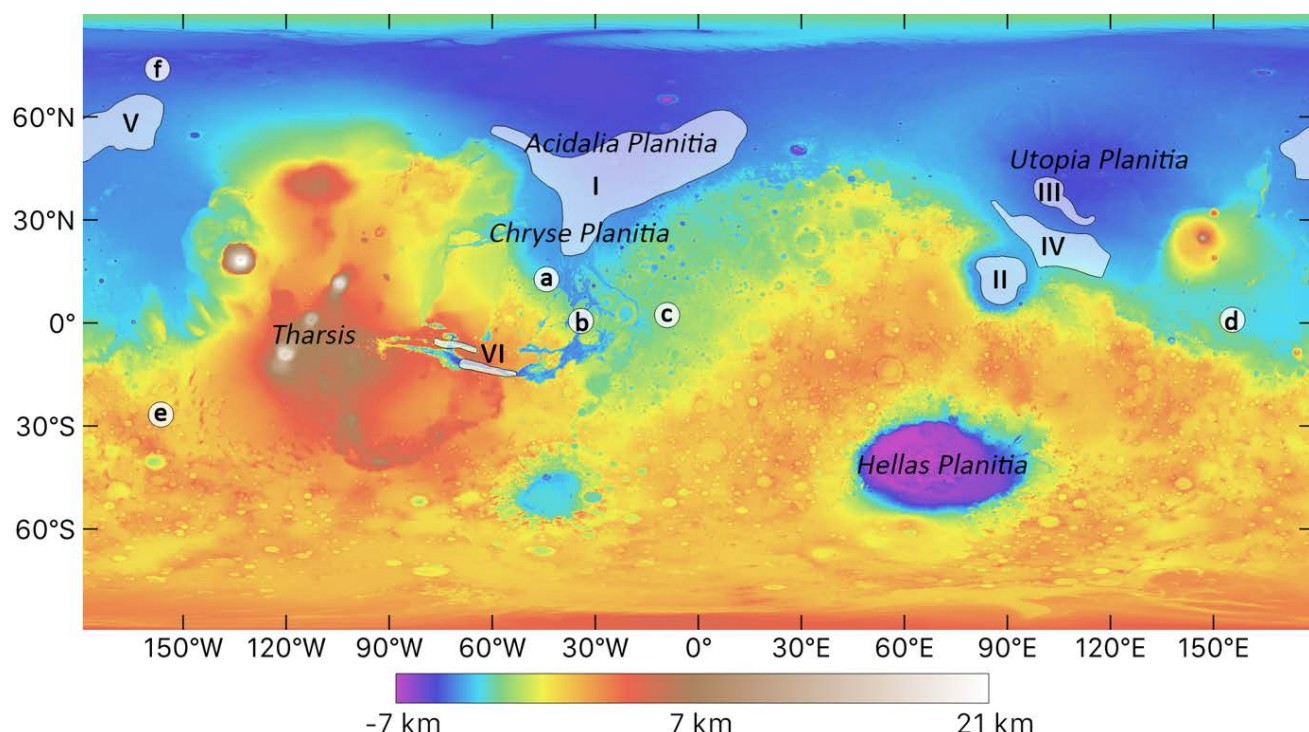

**Figure 1: Global MOLA topographic map showing the positions of the main fields of Martian mud-volcano-like structures. Letters refer to local evidence of sedimentary volcanism in: a) Simud & Tiu Valles, b) Hydraotes Chaos, c)**

**Firsoff crater, d) Medusae Fossae Formation, e) unnamed flat-floored basin in Terra Sirenum, f) Scandia Colles; and roman numerals refer to regional evidence in: I) Chryse and Acidalia Planitiae, II) Isidis Planitia, III) Adamas Labyrinthus, IV) Amenthes/Nephentes region, V) Arcadia Planitia, and VI) Candor and Coprates Chasmata. See the main text for full description of the evidence. MOLA Science Team.**






**Figure 2: A conceptual drawing illustrating the main elements associated with mud volcanism (MV) on Earth both on the surface and in the subsurface. Red dashed line represents fault zone and different colours for arrows mark various sources for liquids. Figure is adapted from Mazzini and Etiope (2017).**



**Figure 3: Sub-kilometre- to Kilometre-scale, circular mounds in Acidalia Planitia.** Panels (a), (c) and (e) are from Mars Reconnaissance Orbiter (MRO) Context Camera (CTX) mosaics (© Google Earth/Mars). Panel (b) is from Mars Odyssey THEMIS-Nighttime infrared (IR) global mosaic, v. 14. (Arizona State University/USGS). Panel (d) is an MRO HiRISE image. Panel (f) is an MRO CTX image. Panel (a), A variety of types of bright mounds. Centred 41.12°N, 26.34°W. Rectangles show locations of Panels (c) - (e). Panel (b), Same area as Panel (a) showing dark responses of the bright mounds and surrounding materials in Nighttime IR. Arrows as in Panel (a). Compare Figs. 3a and b to see the different responses in Nighttime IR of mounds and impact craters. Panel (c), Pitted cones showing central depressions and material on flanks apparently superposed on the darker substrate of the plains. Panel (d), HiRISE image (ESP_026732_2215_RED) showing a bright mound with a central depression and surrounding moat. Panel (e), splotch-like, bright mound with entrained, boulder-sized angular knobs. Panel (f), CTX image (J03_045945_2201_XN_40N027W) showing flow-like features extending from a bright mound (top arrow) to darker apparent flows (lower two arrows). Centred 40.25°N, 27.13°W; location ~62 km SSW of Panel (a). North is up in all. HiRISE imagery: NASA/JPL/University of Arizona, THEMIS imagery NASA/JPL-Caltech/Arizona State University and CTX imagery NASA/JPL/Malin Space Science Systems.






**Figure 4: Examples of kilometre-scale features from various regions of Mars. Panel (a) shows a conical feature with steep flanks and wide central crater (HiRISE ESP_022025_2000, centred 19.73°N, 322.44°E), (b) a domical feature with small central knob in the summit area (HiRISE ESP_025137_1995, centred 19.04°N, 322.64°E), (c) a shield-like or pie-like feature with central breached crater (HiRISE ESP_025704_2005, centred 20.186°N, 321.259°E), (d) a small cluster of conical and domical edifices associated with**

**flow-like unit, and (e) a cluster of wide conical edifices with wide breached central craters surrounded by flow-like unit. Edifices on panels (a,b,c) are situated in the southern part of Chryse Planitia. The fields of cones on panel (d) and (e) are situated on the floor of Coprates Chasma (CTX P13_006269_1670_XN_13S062W, centred 12.711°S, 297.67°E), and in Nepthentes/Amenthes region (CTX G01_018499_1961_XN_16N252W, centred 16.194°N, 107.373°E), respectively. Except the panel (e), north is up. HiRISE imagery: NASA/JPL/University of Arizona and CTX imagery NASA/JPL/Malin Space Science Systems.**




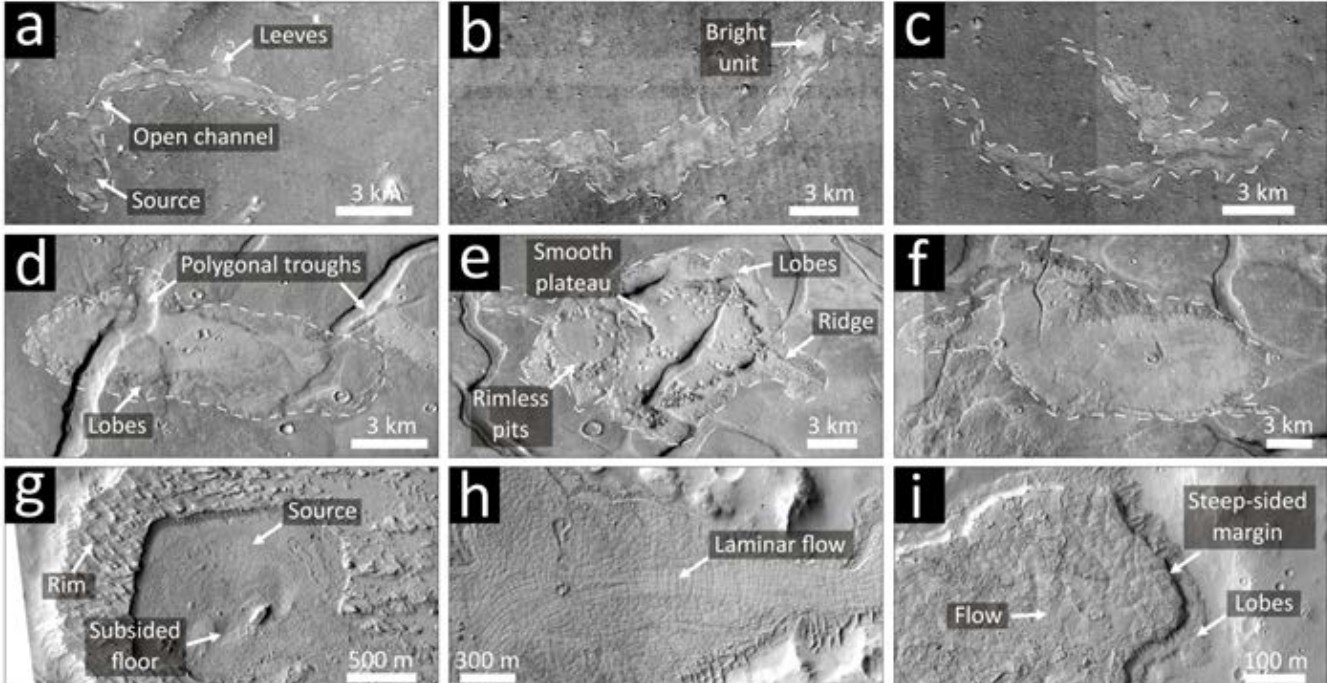

**Figure 5: An example of kilometre-sized flows (KSFs) from three regions of Mars showing large variability in their shapes and general appearance. While the KSFs within Chryse Planitia captured on panels (a,b,c) consist of three different parts – source area, open channel and leeves – and are often associated with bright smooth units, the KSFs in Adamas Labyrinthus (d,e,f) do not show such an appearance. Instead they have the shape of a plateau. Panels (d,e,f) show plateau-shaped KSFs from Adamas Labyrinthus that are standing above the surrounding terrain to a height of 20-30 metres and they do not show clear source area from which the material originated. A unique KSF is the flow of Zephyria Fluctus which shows well-developed large source area (g), as well as former flow pattern on its surface (h) and well-developed steep-sided margins and lobes (i). Panel (a) based on CTX image P17_007639_1997_XN_19N034W (centred 19.85°N, 326.02°E), (b) CTX F01_036121_2011_XN_21N034W (centred 20.41°N, 325.69°E), (c) CTX B19_016856_1990_XI_19N035W (centred 20.228°N, 324.05°E), (d) CTX G21_026424_2175_XN_37N257W (centred 102.2°E, 37.52°N, 102.2°E), (e) CTX P17_007779_2181_XN_38N259W (centred 39.14°N, 100.92°E), (f) CTX P17_007502_2195_XI_39N256W (centred 38.63°N, 104.2°E), (g) HiRISE ESP_037169_1805 (centred 0.59°N, 155.29°E), (h) HiRISE ESP_027464_1805 (centred 0.65°N, 155.39°E), and (i) HiRISE ESP_028941_1810 (centred 0.795°N, 155.592°E) respectively. HiRISE imagery: NASA/JPL/University of Arizona and CTX imagery NASA/JPL/Malin Space Science Systems.**





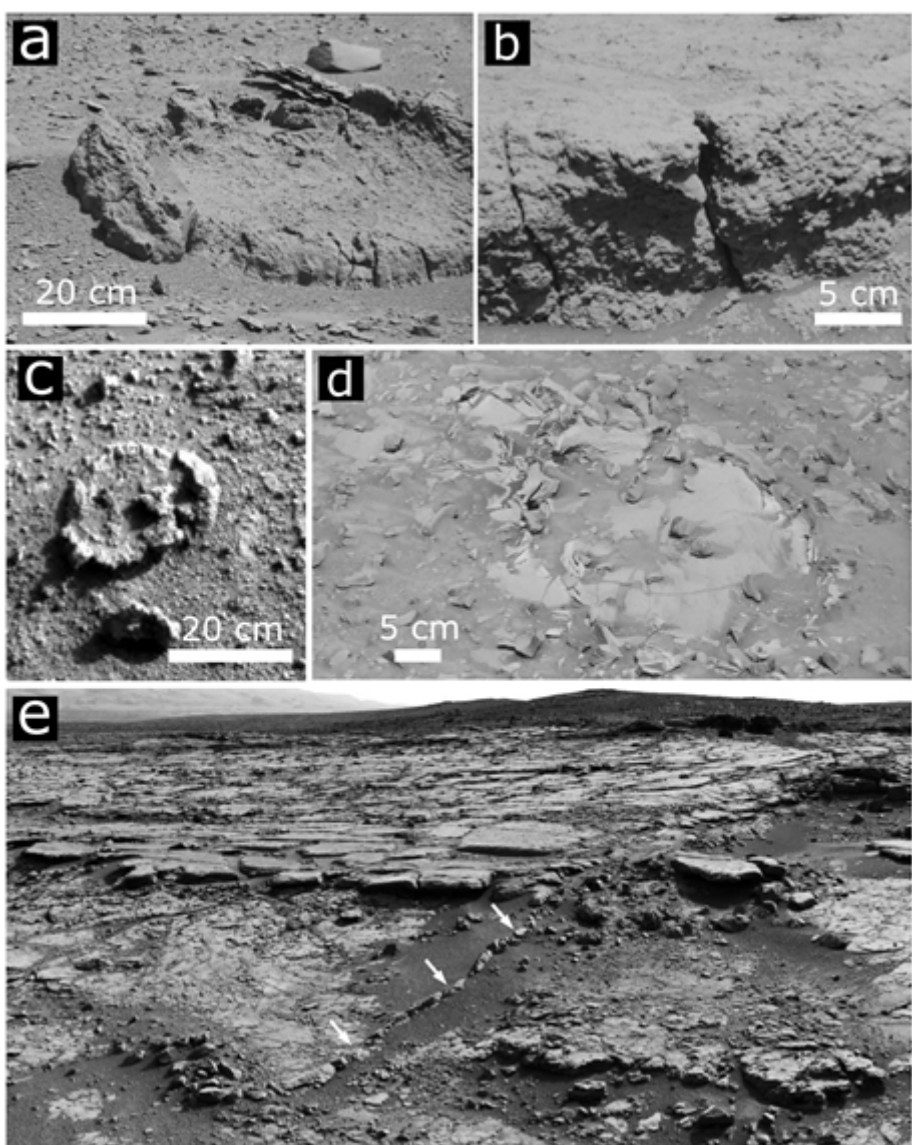

**Figure 6: Examples of outcrop features observed by the Curiosity rover and hypothesised to have formed by upward injection of sediment through the Martian crust. Gale Crater. (a – d). Various pipes observed at Dingo Gap and Marias Pass. Figures are adapted from Rubin et al. (2017). (e). Dike observed at Yellowknife Bay (arrows). Image numbers for (a) 0528MR0020830010303294E01_DXXX, (b) MR002083, (c) NLB_445620806EDR_F0261274NCAM00354M, (d) 1051ML0046250040306086E01_DXXX, and (e) 20170206PIA17595-16. NASA/JPL-Caltech/MSSS.**



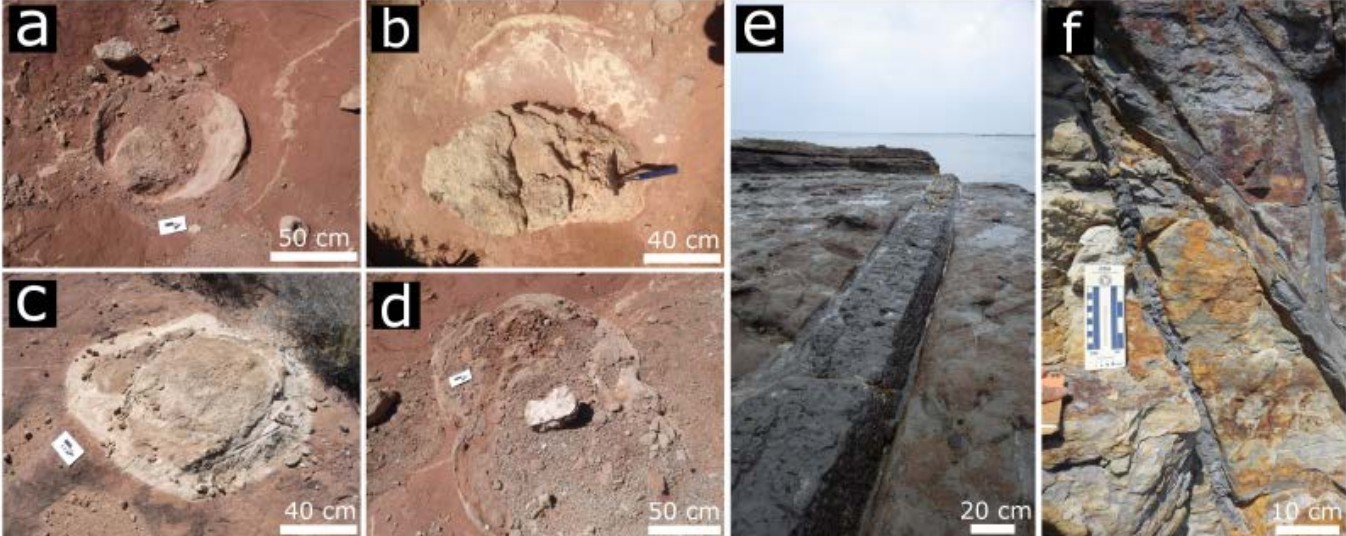

**Figure 7: Terrestrial analogues for the martian outcrop features hypothesised to have formed by upward injection of sediment. (a – d) Clastic pipes widely observed within the Colorado Plateau. (e, f) Sedimentary dikes exposed horizontally (e, the dike width is about 30 cm) and vertically (f) along the western coast of Kii Peninsula. These dikes resulted from an ancient (as old as Miocene) process of subsurface sediment mobilization. Photos in panels (a-d) by David Wheatley (all rights reserved) and (e-f) by Goro Komatsu.**

**Acknowledgment**

We are thankful to David Wheatley for providing us images of clastic pipes in Figure 7 and Dave Rubin for helping us to locate martian images used in Figure 6. AM acknowledge the support of the Research Council of Norway (NFR) through the HOTMUD project number 288299 and its centres of Excellence funding scheme, project number 223272 (CEED). VČ acknowledge the support of IGA Faculty of Environmental Sciences CZU Prague through the grant "Kilometre-Sized Flows on Mars: Controlling Factors and Geomorphology – No. 2022B0038".