# Peer review of "An overview of sedimentary volcanism on Mars"

_EGUsphere, 2022_

## Author Response (AR1)

Reviewer 1

This paper presents an overview of the current state of knowledge with respect to possible mud volcanism on Mars combined with pertinent information from studies of mud volcanoes on Earth. The writing is generally clear and technically accurate, with only a few places where minor clarification is needed. Facts, observations and interpretations are clearly distinguished in the majority of the discussion, with only a few areas where separation of observations and interpretations, or supporting references, are needed. Referencing is generally adequate, although additional references could be added in specific sections to avoid the appearance of speculation. Overall, the paper's strengths outweigh its weaknesses, and the paper would be suitable for publication with minor revision.

> We are thankful to the reviewer for his/her feedback about our work.

Line 85: Provide an explanation for why these two specific questions need to be answered. Are these questions found to be important based on community feedback or through discussions amongst the authors?

> The explanation is within the last sentence of the section "if the answers are yes, the martian mud-volcano-like structures are very likely sites where biomarkers can be better preserved and are, or have been, sites of methane release to the martian atmosphere".

However, we have added the following:

……*we will try to answer two key questions for determining whether the mud-volcano-like structures on Mars may have environmental and biological implications similar to those of terrestrial mud volcanoes:*

Lines 95-115: Clarify as to whether this discussion is based on community consensus or based on the authors' opinions.

> We added the following (with references):

*Natural gas and petroleum geologists agree that the term "mud volcano" should not be used……(....)…….edifices (e.g., Kopf, 2002; Mazzini and Etiope, 2017).*

Line 102: The definition of "mud volcano" being provided here answers one of the "important questions." Is this really a "question"?

> The answer to this comment is already provided in our previous replies. To refrain: the idea here is to verify if characteristics similar to those observed on Earth are also present on Mars addressing some key questions.

Line 116: Provide references to support this statement.

> Mazzini and Etiope (2017, and references therein) is now added, which is the most recent and updated review on mud volcanism on Earth, including discussions on the gas released

Line 187: This doesn't seem to be an enlightening finding since this is the paper's stated definition of a mud volcano.

> The point is about the importance of mud breccia as a carrier of biomarkers. If the martian mud-volcano-like structures are actually mud volcanoes, then they carry biomarkers (proxies of past life) to the surface.

Line 190: Based on this discussion, it doesn't seem that the necessity of gas is in question. This paragraph is really reiterating what is already in the literature regarding the necessity of gas.

> This paragraph is answering the second question, clarifying the importance and the type of the gasses that can be released. From our point of view, it is important that it is in the text.

Line 192: Provide references regarding the necessity of gas in terrestrial mud volcanoes

> Mazzini and Etiope (2017) is the most updated review on terrestrial MV discussing also the role of gas and this reference has been added here.

Line 196: This point is not evident from the previous discussion, but rather it is evident from the literature on terrestrial mud volcanoes. Clarify.

> The text has been modified. Now it reads as follows:

*In conclusion, literature reviews from terrestrial MVs (Mazzini and Etiope 2017) reveal that it is evident that the size of the emission structure, the presence of gas, mud breccia and faults are relevant to understand the genetic process driving mud-fluid manifestations.*

Section 2.4: Rather than calling these key questions, it would be clearer to refer to these as key points, since the answer is already known.

> The answer is not known for a wide community of geologists, especially planetary geologists. In this paper we have reviewed the specific literature to clarify and give direct answers; so we prefer to use the term "key questions".

Line 251: Suggest rewording for the sake of clarification and deconfliction from the next paragraph - "Edifices in these regions display a wide range of shapes (Fig. 4)..."

> Modified as proposed.

Line 294: This might just be a nit-pick, but suggest rewording to clarify, "Based on interpretations of remote sensing data..."

> Modified as proposed.

Line 297: Separate observations from interpretations. Consider something like, "KSFs are often elongated in plan-map view. This elongation is interpreted to be the result of material released from a source area and flowing down the local topographic gradient and filling nearby terrain depressions."

> Modified as proposed.

Line 299: Lower than what? Length to width ratio is not mentioned elsewhere. Elaborate.

> Elaborated, the text now reads as follows: "*However, in flat areas of Chryse and Utopia Planitiae their length-to-width ratio is comparatively lower to those KSFs formed on inclined surfaces.*"

Line 338: Recommend removing Utopia from this sentence since features in Utopia are not discussed in this paragraph.

> Thank you for the suggestion, however, we would like to keep the mention of Utopia here to highlight that mud-volcano-like edifices are also there. However, we modified the next sentence to make it clear that the paragraph is not a comprehensive overview. Now it reads as follows:

*The martian mud-volcano-like surface features have been mainly observed in various parts of the northern lowlands of Mars, especially in Utopia, Chryse and Acidalia Planitiae. For example, in Acidalia, where extensive mapping of more than 18,000 mounds has been completed (Oehler and Allen, 2012a), the circular mounds commonly occur…*

Line 394: A rationale for this statement's implicit assumption that all source reservoirs on Mars should be at a common depth is not given. Either include a more detailed discussion on source depths, or simply state that estimated source depths vary significantly.

> The sentence has been more generalized. Now, it reads as follows: "*It remains unclear at which depth(s) the source reservoir(s) for the hypothesised sedimentary volcanoes is/are located.*"

Line 405: "areally" or "really"?

> The word has been removed completely.

Line 656: Change "in conclusive" to "inconclusive"

> Modified as proposed.

Line 673: Provide pertinent references to terrestrial clastic pipes

> Two references has been added, namely:

Jamtveit, B., Svensen, H., Podladchikov, Y., Planke, S.. Hydrothermal vent complexes associated with sill intrusions in sedimentary basins. Geological Society, London, Special Publications 234, 233-241, 2004.

Svensen, H., Jamtveit, B., Planke, S., Chevallier, L.: Structure and evolution of hydrothermal vent complexes in the Karoo Basin, South Africa. Journal of the Geological Society 163, 671-682, 2006.

Line 700: Remove hyphen from "sub-surface" for consistency with the rest of the paper

> Modified as proposed.

Lines 713-714: Consider updating this statement with an acknowledgment of the current status of this mission.

> The sentence has been updated by inserting the new planned landing date (2031)

Line 723: Here and throughout: Change spelling to "mud-volcano-like" for consistency with the rest of the paper. There are instances of both "mud-volcano like" and "mud volcano-like" variations that need to be corrected.

> Unified across the manuscript.

Lines 739-740: This is an incomplete sentence.

> Fixed. Now the sentence reads as follows:

*Therefore, whether the process of subsurface sediment mobilization on Mars is an exact analogue of the process of terrestrial mud volcanism which is governed by gas emissions remains unsolved.*

Line 749: Consider updating this statement with an acknowledgment of the current status of this mission.

> Sentence was updated.

Line 751: Seems like this sentence belongs in the previous paragraph.

> The pre-last and the last paragraph has been modified to take in consideration the comments above as well as the likely end of the Zhurong mission. New text now reads as follows:

*The Chinese rover Zhurong recently landed near a potential mud-volcano-like mound in Utopia Planitia. This could provide an unprecedented opportunity to collect important observations about the origin of the material(s) forming the edifice. Dedicated future missions (e.g. using drones or helicopters) should focus on sample analysis or even sample collection from some of the potential sedimentary volcanoes in northern latitudes. In parallel, work on analogue experiments through laboratory simulations of mud flows under Mars-like conditions, and associated numerical models will help to assess the origin and significance of some of the various edifices identified on Mars. Future Mars exploration should therefore plan efforts to investigate these enigmatic features, which could be excellent candidates for missions aimed at biosignature detection.*

Reviewer 2

Summary:

This manuscript provides a notable consolidation and discussion of past literature that cite and/or discuss the setting, process, and product of mud volcanism (and similar soft sediment mobilization) on Mars. This paper is concise and straightforward and I encourage publication with some edits, re-phrasings, topical notes, and clarifications.

*> We thank the reviewer for valuable and detailed comments on our manuscript. Our responses are listed below and the manuscript text has been edited to reflect the comments.*

General Comments:

A tabulated representation of all studies that cite mud volcanism or similar might would be very helpful in quickly showing reference, geographic location, reported geometries, and spatial density of identified features. A referenceable table is much preferred to a prose summary of these features and has the potential to shorten the manuscript.

*> A table summarizing specific studies on the manifestations of possible sedimentary volcanism on the surface of Mars has been prepared and is now included in the article. However, we have decided to keep the text at its original length and not to significantly shorten the article.*

Paper does a good of summarizing the features but often leaves out a reference of the data used to make the identification and/or geometric measurements, which is important due to the varied resolution and coverage of such data (and potential missed observational range of features). Update throughout.

*> The manuscript has been updated to include references of the data used.*

Further, the feature summary often implies that there are somewhat strictly bound variances (e.g., line 268 cites "the largest know variation"). The reality is that there is a lateral gradation between many of these fields. That should be noted and perhaps even emphasized.

*> We are thankful for this feedback and we agree with the reviewer that this should be called out clearly. Therefore, we extended the general paragraph at the beginning of the Section 3. Now it reads as follows:*

*"In this section, we briefly summarize the current knowledge about the morphology of mud-volcano-like structures on Mars (Tab. 1). Four individual sub-sections will focus on a) sub-kilometre to kilometre-scale circular mounds widely spread across the northern lowlands, b) the kilometre-scale topographically positive features of various shapes and often associated with flow-like edifices, c) kilometre-scale flows, and d) hundreds of kilometre-long flows and deposits. These divisions were made in order to group features that bear similar morphological, morphometrical and spatial similarities, although overlapping characteristics in some parameters often exist among these groups. Additionally, it should be also noted that edifices when compared among each other within individual fields as well as among different fields*

*often show transition in their shapes. Hence, there is often no strictly bound variance among them and members of different groups can be present in one particular field."*

Preferences in character and origin of mud volcano-like features in certain locations in the manuscript are pronounced, often toward lead author publication. As a review paper, the full range of interpretations should be presented equally, largely because we could all argue, each with success, on the origin of these features. Here, you could even identify where the authors diverge in interpretation and preference.

*> The text has been modified based on the specific comments to reflect this comment. However, we would like to note that we recognize the potential importance of the different types of geologic settings between that of the mounds and Acidalia and Utopia (sedimentary basins with Nighttime IR evidence for fine grained materials and outflow flood history) and the settings for many of the larger, irregular and flow-like morphology features. This latter group of structures may include features with a variety of origins, some of which are likely to be very different from the origins likely to account for the mounds in Acidalia and Utopia, and as such, this was reflected within the manuscript. In the case of larger edifices there is less agreement on the mechanism of their formation than in small ones, where their sedimentary origin is rarely questioned.*

Specific Comments:

Line 40 – The phrasing here (i.e., "never considered") and the following sentence seems to indicate that the process of mud volcanism was considered in passing but not given the level of credence the authors feel was necessary. There is little evidence that the process was considered at all prior to Viking-based investigations. Renewed interest was coupled with high resolution datasets that identified a pervasive population of pitted cones, mounds, and/or associated flows. Please clarify these sentences as such.

*> Modified as proposed. Now the sentence reads as follows: "Despite these observations, mud volcanism was never considered as a process that could have shaped the surface of Mars. Only some early works hypothesised this type of activity based on low-resolution imagery at specific localities where pitted cones, mounds and/or associated flows have been observed (e.g., Davis and Tanaka, 1995; Tanaka, 1997; Ori et al., 2000, 2001)."*

Line 86 – Here, and elsewhere, it is not entirely clear why authors would consider a one-to-one correlation between Earth and Mars. The answers to both these pitched questions could be "No" and "No" and the process of mud volcano-like processes still hold true. That is, on Mars, the material might not originate from "shale" and it might ascend in the absence (or presence) of gas. The phrasing seems to indicate that using the term "mud volcano" in previous planetary literature is erroneous because we do not know if the analogy is exact. However, most (all?) authors that implement the analogy are fully aware of the process differences. Further, though mud volcanism on Earth might originate in environments that contain, preserve, and mobilize biogenic lithologies, it is to be debated whether such eruption on Mars is tapping the same kinds of material. Some of these questions are targeted in this

paper, but the narrowness of these two questions and the absolutism that the authors link them should be lightened slightly.

*> We have now re-phrased this section and, as advised, we have lightened up our statements. We clarified that if certain prerequisites that are well-defined to classify mud volcanism of Earth are also fulfilled on Mars, then the martian mud-volcano-like structures may have environmental and biological implications similar to those of terrestrial mud volcanoes.*

Line 102 – Same as above comment. The direct link to "shale" is problematic in this paper. This may be the case on Earth, as argued by the authors, but is not necessarily true or even preferred on Mars. The discussion of semantics is important and spot on. However, planetary literature specifically hedges the citation of "mud volcanism" to emphasize that the direct, one-to-one similarity to Earth is unknown. The paper, thus, seems to indicate that the use of "mud volcano" on Mars must have the same critical characteristics, else the term is incorrect. I take a broader approach that when authors use "mud volcanism" on Mars, they are almost inherently identifying differences. This sematic issue is not unique to mud volcanism but almost every process that we apply, as terrestrial geologists, to other bodies. Clarify the discussion of semantics or offer a broader statement that allows for application and adaptation of the term to Mars. Section 2.1 needs a discussion about the latitude afforded for places we cannot directly observe.

*> In this section we provide a summarized description of the main characteristics used to define mud volcanism on Earth. These are essential key aspects to provide the right classification for this geological phenomenon. Note that in this section (S 2.1) we do not discuss the morphological features observed on Mars.*

Line 108 – Replace "ground up" with "comminuted".

*> Modified as proposed.*

Line 177 – Again, I do not find these two questions to be all that significant in elucidating the process of mobilization and ascent of muddy and/or breccia-rich fluid on Mars. The "answer" could be that neither of these are true and, yet, the process still operates on Mars. As such, the discussion up to this point seems to encourage the reader to abandon the use of "mud volcano" as a term of comparison and reference for Mars-observed features. It seems that if these two things are not true, then the authors do not appreciate the application of the term…this paper then becomes more of a discussion of semantics wherein alternative terms should be offered up.

*> The initial part of the manuscript is designed to guide the reader throw a logical reasoning regarding a) key parameters used to classify mud volcanism on Earth (S 2.1, 2.3); b) provide examples of various surface phenomena on Earth that are similar to mud volcanism but that are erroneously classified as such (S 2.2); c) verify that the key parameters are fulfilled on known mud volcanoes on Earth (S 2.4); d) convey the message that if the known terrestrial characteristics are not present/known on martian features, then ambiguity remains and that regardless these structures may have relevant environmental and biological implications.*

*Several authors (including co-authors of this manuscript) have previously classified some martian structures as "mud volcanoes" based largely on morphological observations. The same mistake often*

*happens on Earth and several structures are considered as mud volcanoes even when this is not the case (see detailed examples in section 2.2). Therefore, given the many uncertainties, we would like to stress the need of using the terminology of "mud-volcano-like" instead when observing other celestial bodies. This, we believe, is a key message for the manuscript.*

Line 182 – "Typically originates from deep shale deposits" seems to be at odds with preceding information wherein the authors very directly and absolutely tie the process of buried shale. Clarify if shale is a requirement or just a possibility on Earth.

*> We don't understand this comment. As stated in the text, the presence of shale, its buoyancy and extrusion are "essential components" of MVism.*

Line 204 – The use of "mud volcano-like" here seems to provide an answer to the semantic question posed by the authors. That is, "please don't call these features mud volcanoes on Mars because we (the scientific community) do not have the knowledge about the shale origin or association with gas." Again, I find that the preceding sections would benefit from simply acknowledging more directly that a strict application of "mud volcanism" needs to account for significant potential unknowns.

*> We have now clarified this concept as suggeste*d.

Line 215 – This citation list might benefit by referencing Viking-based geologic maps by Scott and Tanaka.

*> Reference has been added.*

Line 226 – Please cite the data that was used and, thus, help readers understand the lower limit of the measurements … there could be more features beyond the resolution of those identified and that is note-worthy.

*> Modified as proposed.*

Line 229 – Here and elsewhere, please cite the instrument that provided the image and/or topography data.

*> Modified as proposed.*

Line 232 – Please cite the quantify that indicates "many" which appear as domes.

*> Modified as proposed.*

Line 235 – Farrand was not the first to recognize these as domes, cones, and splotches. Scott et al. geologic mapping (USGS I-1802) described these as such. Also, see Scott and Underwood "Mottled Terrain: A continuing Martian enigma". Delete or re-phrase this line.

*> We are thankful for catching this mistake. The text has been rephrased. Now it reads as follows: "These types of mounds were described as early as 1986 by Scott and Tanaka and in 1991 by Scott and Underwood. Farrand et al. (2005) described these morphologies as domes, cones, and splotches."*

Line 245 – Not clear or demonstrated that there is relative uniformity in the fields of cones across the lowlands. What do the authors consider to be "uniformity"? Also, the "km-scale feature type" should include those identified by Oehler and others … as cited on the preceding page, these features range from 0.3 to 2.2 km. They are, thus, kilometer-scale. Re-phrase and clarify.

*> This section has been reworked to better clarify the reasoning why described edifices are considered not to share genetic origin with sub-kilometre to kilometre-scale circular mounds described in previous section.*

Line 250 – Correct spelling of "Nephentes" to "Nepenthes" here and throughout.

*> Modified as proposed.*

Line 267 – Please indicate how much below the floor of surrounding plain, and what proportion are observed as thus.

*> The text has been clarified. Now it reads as follows: "The crater floors can be at or a few dozens of meters below the preeruptive level of the surrounding plains (observed for 13 edifices out of 47 measured, for details see Table S.1 in the auxiliary material in Brož and Hauber, 2013)."*

Line 269 – Replace "can be" with "have been".

*> Modified as proposed.*

Line 274 – The strong interpretive preference of lead author on these larger features is clear here, though the origin of these features are no more or less "settled' than any other field. In fact, it has been argued (by me) that the physiographic location of some of these features may imply a sedimentary origin (for which some of these authors disagree, which is their right). The issue here is that you should provide alternative hypotheses across all the features described herein, including the relatively smaller features identified throughout the lowlands. Selective citation of alternatives is not helpful in such a review paper.

*> The text has been shortened, now it reads as follows: "Because they have a similar variety of shapes and are commonly associated with flows, it is common for terrestrial km-scale mud volcanoes to be proposed as potential analogues (e.g., Jakubov et al., 1971; Aliyev et al., 2015; Mazzini and Etiope, 2017). However, igneous volcanism has been invoked as an alternative mechanism for their formation in several cases and the debate about their formation mechanism is not yet settled (Lucchitta, 1990; Meresse et al., 2008; Brož and Hauber, 2013; Brož et al., 2015a; 2015b; 2017)."*

Line 285 – Thought the end of the paragraph strikes the right tone (i.e., "spectral data is not all that helpful"), it comes a little late. Preceding lines cite spectral details to isolate a preferred interpretation (juvenile magmatic volcanism). Spectral data for Mars is almost always going to show basaltic mineralogizes with various indications of water-related alteration.

*> The paragraph has been shortened to make clearer that spectral data do not support one formation mechanism over another. Now it reads as follows: "The spectral data from CRISM have been used to gain insight into their formation mechanism, however, to date, spectral observations do not support a*

*sedimentary origin over an igneous one, or vice versa. This is because data did not unambiguously show the presence of phyllosilicates, carbonates, or sulphates in association with these edifices (Dapremont and Wray, 2021; see also Brož et al., 2017 for additional discussion).*

Line 294 – Not sure "most likely" is the best choice of words here. Perhaps "have also been interpreted as the result of either juvenile magmatic or mud volcanism" or similar.

*> "Most likely" has been replaced by "plausible"*

Lines 303 to 304 – The line "has been interpreted to be formed by the emplacement of mud … however, its igneous origan was not definitely ruled out". This seems to be the case for all the features described in this section. Thus, it is unclear why this one flow was specifically called out.

*> Removed.*

Line 311 – There is no Skinner et al. (2009) in the reference list.

*> Corrected to Skinner and Mazzini (2009).*

Line 327 – Replace with "which they interpreted as indicative of a relatively finer grain size".

*> Modified as proposed.*

Line 329 – Here, again, is phrasing that skews the interpretations toward a volcanic origin. These discrepancies in interpretation (e.g., high v. low thermal inertia) can be interpreted multiple ways. Author preferences are exposed in these kinds of statements, which somewhat undercuts the value of a review paper.

*> The sentence was deleted.*

Line 331 to 335 – These lines can be deleted. There are multiple large, lobate units that occur on Mars, which have been interpreted as mud or lava. This does not add anything to the discussion.

*> Removed as proposed.*

Line 347 – Remove "facies" in "distal facies sediments".

*> We would rather leave in "facies" - for clarity for those who might not be familiar with the concept and might be perplexed by "distal sediments.", but we have, however, made distal-facies hyphenated to improve the phrase.*

Line 357 – Capitalize "Late" (and Early and Middle) when referring to ages, here and throughout, even when Tanaka et al. (confusingly) referred to the units using a lower case letter.

*> Modified as proposed.*

Line 405 – Define "really extensive" or change the phrasing. Also, might be good to note that absolute and relative age dating is problematic in general and, very specifically, on the surfaces that contain these

features of note.

> *Modified to kilometre-scale flows and the end of this section has been supplemented by the following: "However, it should be noted that remote sensing methods based on absolute or relative techniques are still not entirely accurate tools for determining age, especially for small-scale buildings. Readers should therefore approach the above data with caution."*

Line 409 to 410 – Comma splice.

> *Modified as proposed.*

Line 502 – Process interpretation for mud volcanism on Mars suffers no more of less than other interpreted geologic processes. The discussion here seems to indicate that the mud volcano process is particularly difficult to assess, when it actuality the validity of most (all?) processes are equally difficult to assess.

> *The intention of this sentence was not to suggest that this process is particularly difficult to assess, but to point out that we lack a basic theoretical understanding of how the process of sedimentary volcanism should occur on Mars, and therefore to motivate the reader to study this aspect in detail. However, the sentence was eventually removed, as this is also clear from the conclusion of the previous paragraph.*

Line 515 – Not sure how the assessment outlined herein resolves that sediments should be accumulated quickly in all case or even what "quickly" means. Also, it has not been demonstrated that gas is a critical component of this process on Mars. I think there is a case to be made for Martian strata to be "undercompacted" and particularly susceptible to both slow and rapid compaction (see the number of irregularly shaped depressions in the northern plains), perhaps associated with ascent of mobilized sediment. Could cite as much here.

> *Thank you for this comment, which made us think twice about sticking too closely to terrestrial analogy. Actually, our wording here was governed by analogy to terrestrial mud volcanism and previous studies which assumed rapid compaction (without discussion what "rapid" actually means; e.g., Di Pietro et al., Icarus, 2021), but indeed sediment mobilization and expulsion on Mars may not necessarily have occurred under the same conditions as it does on Earth. Nevertheless, a few considerations point to a "rapid" accumulation (and subsequent compaction) of massive amounts of sediments in the northern lowlands, where most mud-volcano-like landforms are observed. There is a general consensus that the floods that carved the outflow channels were formed in a "catastrophic" way, i.e. with very high discharge rates and at correspondingly short timescales. Therefore, these floods transported large amounts of volatile-rich sediments into the northern lowlands, where they accumulated as thick sediment deposits most likely in a geologically short ("rapid") time. Thus, it seems reasonable to expect rapid sediment accumulation and compaction at some of the major candidate sites for sedimentary volcanism (Acidalia, Utopia), although it would admittedly be somewhat speculative to estimate specific sedimentation and/or compaction rates (there is very little published literature on sediment compaction on Mars; for one of these rare studies see Gabasova and Kite, 2018) . In a similar sense, the involvement of gas may have been a plausible factor in sedimentary volcanism on Mars.If organics were buried by the outflow channel sediments, the thick sediment deposits (several kilometers) together with the geothermal gradient at these times may have led to thermogenesis of methane (see discussion by Oehler*

*and Etiope, 2021). Anyway, we agree with the reviewer that neither the rapid sediment accumulation nor the role of gas are confirmed by observations as prerequisites for martian sedimentary volcanoes to form.*

*We therefore added one sentence for each of the two unknowns (rapid sedimentation/compaction and the role of gas) at the end of the specific paragraphs, containing more cautious wording regarding the role of rapid accumulation/compaction and gas as necessary prerequisites for mud-volcano like structure formation.*

Line 525 – Might want to differentiate that there are potentially multiple environments within and around muti-ring impact basins…center and circumferential annular spaces.

*> The text has been modified to take this into account. Now it reads as follows: "...ancient, large impact basins that would have acted as sinks for water and sediments that were transported in giant flood events through outflow channels (e.g., McGill, 1989; Frey et al., 2002; Ivanov et al., 2014; Jones et al., 2016) together with their circumferential annular spaces (e.g. Skinner and Tanaka, 2007), and…"*

Line 548 – Might want to differentiate seismicity between quakes and impact-induced seismicity.

*> Modified as proposed. Now it reads as follows: "seismic activity caused by internal martian processes or by impacts "*

Line 560 – This is an interpretation of this paper (i.e., more activity in Noachian and Hesperian). Noachian age features may not exist. Also, though statistics show larger impact craters to be less frequent in the Amazonian, there is still a whole lot of time for larger impacts. From what we know right now, many of these features form on or within Late Hesperian and Early Amazonian units.

*> Again, we thank the reviewer for this hint. We completely agree, and have modified the manuscript to reflect the fact that the sediment mobilization may have occurred throughout martian history, but that some potential triggers (such as the thickening cryosphere) may have been relatively more important in the Hesperian and Amazonian as compared to the Noachian, which may explain that the observed mud-volcano like structures are not Noachian in age. We note here, however, that this observation does not imply that mud-volcano like structures did not form in the Noachian: Potential Noachian mud-volcano like structures may have been buried by later sediments, or they may have been eroded as they are susceptible to water and wind erosion due to their fine-grained nature and (probably) poor consolidation.*

Line 656 – Definitive proof of a great many processes are not achievable for Mars. This phrasing, again, seems to come across as the mud volcano interpretation is particularly susceptible to definitive determination. It is not any more than other processes.

*> Thank you for the comment. We fully agree with the comment in saying that definitive proofs for a great many processes are not achievable for Mars. We rephrased sentences slightly and the new test reads as follows:*

*"Currently available remote sensing data, despite their diversity and high spatial resolution, provide only limited insights into the nature of the proposed candidate sedimentary volcanoes. The origin of a specific*

*landform can be interpreted differently because of equifinality, a problem planetary geology suffers in investigation of many surface features (e.g., Beven, 1996; Komatsu, 2007). Spectral data obtained from orbit are not conclusive, either (e.g., Komatsu et al., 2016; Dapremont and Wray, 2021). A definitive proof of sedimentary volcanism on Mars is hampered by the lack of in-situ data (ground truth) that could provide unambiguous evidence of their formation. Until now, in situ observations supporting the presence of subsurface sediment mobilization are very limited. In this chapter, we describe the most promising examples that have been, or can be, visited on the martian surface, and suggest tests of the sedimentary volcano hypothesis."*

Section 7 – I am also not convinced that this section is needed. Sites to visit and observe are broad. These features have not been directly observed. The Gale example is not particularly relevant to the discussion herein (it was never really brought up before). These kinds of "what ifs" could easily be conveyed in shortened format in a Conclusion section.

*> Thank you for the comment. We believe that this section is necessary for the paper because of the following reasons:*

*This paper is a review of sedimentary volcanism. And it intends to summarize relevant observations, both orbital remote sensing and in-situ investigations that have been historically presented. We think that the future of sedimentary volcanism study on Mars is in-situ investigation. Despite all the great efforts of many studies, past attempts to confirm the sedimentary volcanism on Mars using orbital remote sensing have failed. Without proper in-situ investigation of candidate sedimentary volcanism sites, it is likely that we will not advance much further in this science. So, referring to in-situ investigations in this section is quite appropriate.*

*Sedimentary volcanism is not only what we see on the surface. The subsurface process is as important as what it constructs on the surface. In this sense, identifying features such as sedimentary dikes and diapirs can give important information on how sediment/water mixture moved in the subsurface environment. The Gale example found with in-situ investigation is quite interesting in this sense because it indicates past subsurface movement of a sediment-liquid mixture. Whether it is a type of clastic pipes common on the Colorado Plateau or mud dikes related to classic terrestrial mud volcanoes, we don't know. However, since we still don't understand how sedimentary volcanism works on Mars since we have not confirmed its presence, having an open mind is important not to exclude potentially relevant phenomena.*

*We understand that the Zhurong rover mission is close to its end. However, its investigation apparently sparked strong interest in China for mud volcano study as evidenced by a number of publications in recent years, many of which are interpreting pitted cones in the landing site area as potential sedimentary volcanoes (they call them mud volcanoes). It's the only landing mission which has approached a field of potential sedimentary volcano edifices, and for this reason this event is historically important for sedimentary volcanism study on Mars. So, we would like to keep it in the paper.*

*In a broad sense, all the above elements included in Section 7 can be part of a story of studies regarding what we call sedimentary volcanism on Mars. It should also suggest future directions and we also did so in this section. Overall, for the context of a proper review on sedimentary volcanism study on Mars, this section is important and needed.*

Line 739 – I think it is probably most obvious and tractable to say that it is unlikely that there is a one-to-one connection between mud volcanism on Earth and on Mars. Mud volcanism on Earth has very particular constraints, as documented herein. However, we do not know whether those constraints exist on Mars. This does not preclude us from using the mud volcano term. It is broadly recognized that these processes are not likely to be truly equivalent.

> *We have now clarified and summarized our conclusions as suggested by the reviewer.*